# Mitotically heritable, RNA polymerase II-independent H3K4 dimethylation stimulates *INO1* transcriptional memory

**Bethany Sump, Donna G Brickner, Agustina D'Urso†, Seo Hyun Kim‡, Jason H Brickner\***

Department of Molecular Biosciences, Northwestern University, Evanston, United States

**Abstract** For some inducible genes, the rate and molecular mechanism of transcriptional activation depend on the prior experiences of the cell. This phenomenon, called epigenetic transcriptional memory, accelerates reactivation, and requires both changes in chromatin structure and recruitment of poised RNA polymerase II (RNAPII) to the promoter. Memory of inositol starvation in budding yeast involves a positive feedback loop between transcription factor-dependent interaction with the nuclear pore complex and histone H3 lysine 4 dimethylation (H3K4me2). While H3K4me2 is essential for recruitment of RNAPII and faster reactivation, RNAPII is not required for H3K4me2. Unlike RNAPII-dependent H3K4me2 associated with transcription, RNAPII-independent H3K4me2 requires Nup100, SET3C, the Leo1 subunit of the Paf1 complex and, upon degradation of an essential transcription factor, is inherited through multiple cell cycles. The writer of this mark (COMPASS) physically interacts with the potential reader (SET3C), suggesting a molecular mechanism for the spreading and re-incorporation of H3K4me2 following DNA replication.

**\*For correspondence:**
j-brickner@northwestern.edu

**Present address:** †Arcturus Therapeutics, San Diego, California, United States; ‡Virginia Tech Carilion School of Medicine, Roanoke, Virginia, United States

## Editor's evaluation

The findings in this report are highly significant in providing evidence that a positive feedback loop exists between Sfl1-dependent interaction with the nuclear pore complex and H3K4me2 deposition in the INO1 promoter, which is essential for recruitment of poised Pol II, but which does not require transcription initiation to occur. This distinguishes this activity of COMPASS from its conventional role in H3K4 methylation that is dependent on transcription. The Pol II-independent mechanism was also shown to require Nup100, SET3C, and the Leo1 subunit of the Paf1 complex. It is further noteworthy that this specialized H3K4me2 deposition can persist through multiple cell cycles, in a manner that appears to be enhanced by physical association between the writer of this mark, the memory-specific form of COMPASS lacking subunit Spp1, and the reader SET3C, in the manner expected for epigenetic spreading of histone methylation.

## Introduction

Cells react to changes in their environment by altering gene expression, primarily by regulating transcription of inducible genes. The rate of induction of such genes is a product of enhancer and promoter activity (*Vo Ngoc et al., 2017*) but can also be influenced by the previous experiences of the cells. A number of genes from yeast, flies, worms, mammals, and plants are more strongly induced following a previous exposure to a stimulus, and this primed state can persist for 4–14 mitotic cell divisions (*Brickner et al., 2007*; *D'Urso et al., 2016*; *Gialitakis et al., 2010*; *Lämke and Bäurle, 2017*; *Light et al., 2013*; *Light et al., 2010*; *Maxwell et al., 2014*; *Pascual-Garcia et al., 2017*; *Siwek*

*et al., 2020*; *Sood and Brickner, 2017*). Because only a subset of genes induced under a particular condition exhibits memory (*Light et al., 2013*), this can also result in a qualitative change in the global expression pattern.

Although transcriptional memory in different organisms or for different genes has unique features, several common, conserved mechanisms have been identified. For example, in budding yeast, flies, and mammals, transcriptional memory requires the nuclear pore protein Nup98 (homologous to Nup100 in yeast). This protein physically interacts with the promoters of genes that exhibit memory and loss of Nup98/Nup100 disrupts memory (*Light et al., 2013*; *Light et al., 2010*; *Pascual-Garcia et al., 2017*). The interaction with Nup98 with chromatin in flies and mammals can occur away from the nuclear pore complex (NPC; *Capelson et al., 2010*; *Kalverda et al., 2010*). However, during memory, the interaction of promoters with Nup98/Nup100 in both yeast and flies occurs at the pore.

Transcriptional memory is also associated with local changes in chromatin modifications and chromosome folding. In *Drosophila*, memory induced by ecdysone involves a long-distance promoter-enhancer interaction that is strengthened/stabilized by Nup98 (*Pascual-Garcia et al., 2017*). In yeast, plants, and human cells, histone modifications of the promoter are required for memory: histone H3 lysine 4 dimethylation(H3K4me2) is generally associated with memory in yeast and humans (*D'Urso et al., 2016*; *Gialitakis et al., 2010*; *Light et al., 2013*; *Light et al., 2010*), while in plants, several forms of epigenetic memory are associated with either H3K4me2 or H3K4me3 (*Lämke and Bäurle, 2017*). In flies and mammals, Nup98 physically interacts with the H3K4 methyltransferases Trx and Set1A/COMPASS, respectively (*Franks et al., 2017*; *Pascual-Garcia et al., 2014*). Nup98/Nup100 is required for memory-associated H3K4me2 in yeast and mammals, and conditional inactivation of either COMPASS (the methyltransferase) or SET3C (a histone deacetylase complex that binds H3K4me2; *Kim and Buratowski, 2009*) leads to rapid loss of memory in yeast (*D'Urso et al., 2016*; *Light et al., 2013*; *Light et al., 2010*). Likewise, substitution of alanine or arginine for lysine 4 on histone H3 (H3 K4R or K4A) disrupts *INO1* transcriptional memory (*D'Urso et al., 2016*). Importantly, the H3K4me2 during memory in yeast is carried out by a form of COMPASS that lacks the Spp1 subunit (Spp1⁻ COMPASS), preventing trimethylation (*D'Urso et al., 2016*).

Finally, transcriptional memory in yeast and mammals leads to binding of a poised RNA polymerase II (RNAPII) pre-initiation complex (PIC; *D'Urso et al., 2016*; *Light et al., 2013*; *Light et al., 2010*). A similar phenomenon, called RNAPII 'docking,' has been reported in *Caenorhabditis elegans* (*Maxwell et al., 2014*). Work from yeast suggests that this poised RNAPII PIC is distinct from active RNAPII PIC in two ways. First, it fails to recruit Cdk7 (Kin28 in budding yeast), the kinase that phosphorylates serine 5 on the RNAPII carboxy terminal domain upon initiation (*D'Urso et al., 2016*). Second, it remains associated with Mediator kinase Cdk8 (Ssn3 in budding yeast; *D'Urso et al., 2016*). Cdk8 is also associated with poised promoters in HeLa cells (*D'Urso et al., 2016*) and has been found to regulate initiation (*Akoulitchev et al., 2000*; *Pavri et al., 2005*), suggesting that it plays a conserved role in transcriptional poising. Conditional depletion of yeast Ssn3 from the nucleus leads to loss RNAPII binding from the *INO1* promoter during memory and a defect in the rate of *INO1* reactivation (*D'Urso et al., 2016*). Therefore, transcriptional memory is a conserved phenomenon that may involve a conserved core mechanism.

The highly inducible budding yeast gene *INO1* has served as a model for epigenetic transcriptional memory. *INO1* is an essential enzyme that catalyzes the conversion of glucose to *myo*-inositol for the biosynthesis of phosphatidylinositol. Our current understanding of *INO1* memory is summarized in *Figure 1A*. Repressed *INO1* localizes in the nucleoplasm. Upon activation, the gene repositions to the nuclear periphery through interaction with the NPC (*Ahmed et al., 2010*; *Brickner and Walter, 2004*; *Sumner et al., 2021*). Interaction of active *INO1* with the NPC requires two transcription factors, Put3 and Cbf1, that bind to the upstream DNA zip codes GRSI and GRSII (*Figure 1A*; *Ahmed et al., 2010*; *Brickner and Walter, 2004*; *Randise-Hinchliff et al., 2016*). Upon repression, recently repressed *INO1* acquires memory-specific chromatin marks, poised RNAPII PIC, and associates with the NPC by another mechanism (*Brickner et al., 2007*; *Light et al., 2013*; *Light et al., 2010*). Whereas nucleosomes in the promoter and 5′ end of active *INO1* are hyper-acetylated and show both H3K4me2 and H3K4me3; the nucleosomes at the 5′ end of recently repressed *INO1* are hypo-acetylated and show only H3K4me2 (*D'Urso et al., 2016*; *Light et al., 2013*). Also, the histone variant H2A.Z is incorporated into an upstream nucleosome during *INO1* memory (*Brickner et al., 2007*; *Light et al., 2010*). The interaction with the NPC requires the Sfl1 transcription factor, the memory recruitment sequence

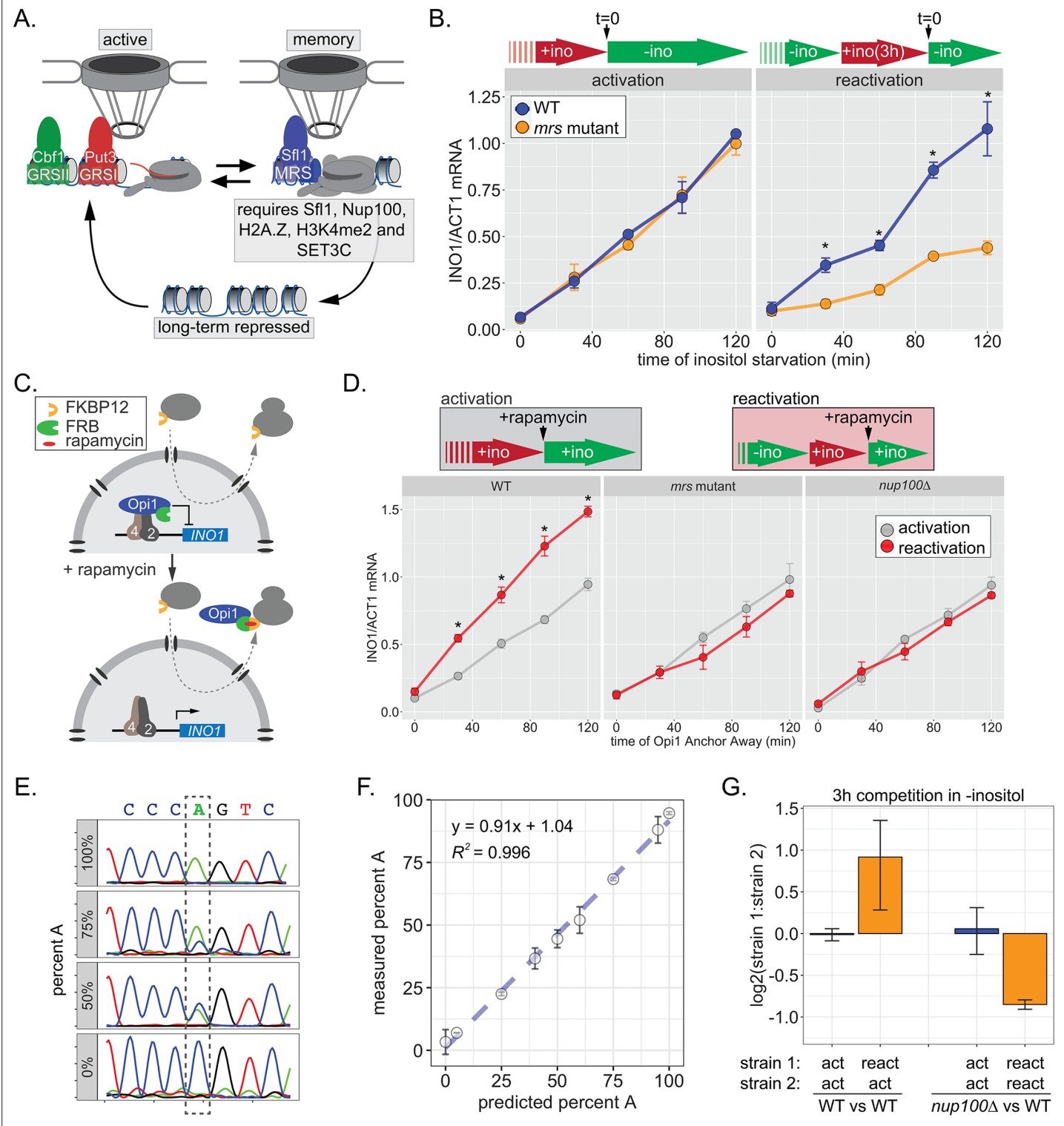

**Figure 1.** *INO1* transcriptional memory stimulates faster transcription and provides a competitive fitness advantage. (**A**) Model of *INO1* in the active, memory, and long-term repressed states, highlighting factors that are specifically required for memory. (**B**) Activation (left) and reactivation (right) of *INO1* in wild type (WT) and *mrs* mutant strains upon starvation of inositol. Cells were harvested at indicated time points and the *INO1* mRNA was quantified relative to *ACT1* mRNA by real time quantitative PCR (RT-qPCR) (*p-value<0.05 from one-tailed t-test comparing WT and *mrs* mutant, alternative = greater). (**C**) Schematic of Anchor Away of Opi1 to induce *INO1*. (**D**) Top: experimental scheme for synthetic activation and reactivation of *INO1*. Activation and reactivation of *INO1* in WT (left), *mrs* mutant (middle), and *nup100Δ* (right) strains upon removal of Opi1 by Anchor Away in the presence of inositol. Bottom: Cells were harvested at indicated time points and *INO1* mRNA was quantified relative to *ACT1* mRNA by RT-qPCR (*p-

*Figure 1 continued on next page*

*Figure 1 continued*

value<0.05 from one-tailed t-test comparing reactivation and activation, alternative = greater). (**E**) Chromatograms resulting from sequencing mixtures of strains having either 'A' or 'C' SNP within an integrated plasmid, as indicated (dashed box). (**F**) Standard curve comparing the predicted percentage of strain A (as estimated by O.D.$_{600}$) with the measured percentage of A (as quantified by the relative area under the peaks, as shown in E). (**G**) Relative abundance of competing strains, as indicated. The log$_2$ ratio of the abundance of the two strains after 3 hr of competition in media lacking inositol is shown. For panels B, D, F, and G, data are averages of three biological replicates ± SEM.

The online version of this article includes the following figure supplement(s) for figure 1:

**Figure supplement 1.** *CHO1* exhibits inositol transcriptional memory.

**Figure supplement 2.** *INO1* memory does not require transcription.

(MRS) DNA zip code to which Sfl1 binds, and the nuclear pore protein Nup100 (*D'Urso et al., 2016*; *Light et al., 2013*; *Light et al., 2010*). Loss of Nup100 or Sfl1, or mutations in the MRS, disrupt *INO1* localization to the nuclear periphery, cause loss of H3K4me2, H2A.Z and RNAPII binding, and slow the rate of reactivation of *INO1* without affecting repression or activation. *INO1* memory, as monitored by gene positioning, histone modification or RNAPII binding, is both maintained in mother cells and inherited to daughter cells through up to four mitotic divisions before being lost (*Brickner et al., 2007*; *D'Urso et al., 2016*; *Light et al., 2010*).

Here we exploited our knowledge of *INO1* transcriptional memory to address four critical questions. First, although transcription rates are impacted by memory, how it impacts fitness has not been generally assessed. We find that inositol memory provides a competitive fitness advantage during inositol starvation that is Nup100-, Cdk8-, and H3K4me2-dependent. Second, we assessed the function of H3K4me2 during memory. Sfl1 is required for H3K4 dimethylation, and H3K4me2 is essential for both Sfl1 binding and for RNAPII recruitment during *INO1* memory, suggesting that memory involves a chromatin-dependent positive feedback loop. Third, we define ways in which the dimethylation of H3K4 during transcription is different from dimethylation of H3K4 during memory. While transcription-associated H3K4 methylation is RNAPII-dependent, memory-associated H3K4me2 is RNAPII-independent and requires both overlapping and distinct factors. Finally, we explore the molecular mechanism of epigenetic inheritance of *INO1* memory. We rule out that protein production during inositol starvation promotes memory and instead highlight the critical role of heritable H3K4me2. Although establishing H3K4me2 during memory requires Sfl1, once established, H3K4me2 is maintained and inherited through ~4 cell divisions in the absence of Sfl1. A putative reader of H3K4me2 (SET3C) that is essential for this inheritance physically interacts with the writer of H3K4me2 (COMPASS), suggesting a molecular mechanism by which H3K4me2 is inherited during DNA replication. This work provides a compelling example of a heritable histone modification that stimulates future transcription and is inherited over a shorter timescale than other heritable histone modifications like H3K9 or H3K27 methylation.

## Results

### Epigenetic transcriptional memory of inositol starvation stimulates faster transcription and promotes competitive fitness

Factors that are specifically required for *INO1* memory stimulate faster reactivation. For example, while the rate of *INO1* activation (+inositol → −inositol) is unaffected by mutations in the MRS, the rate of *INO1* reactivation (−inositol →+inositol, 3 hr → −inositol) is clearly decreased by such mutations (*Figure 1B*). Thus, Sfl1 binding to the MRS is important for enhancing the rate of *INO1* reactivation and has no role in *INO1* activation. However, this experiment highlights a paradox: although poised RNAPII PIC is associated with the recently repressed *INO1* promoter prior to reactivation (*D'Urso et al., 2016*; *Light et al., 2010*), the rate of reactivation is not obviously faster than the rate of activation (*Figure 1B*). This has led to the suggestion that *INO1* does not exhibit transcriptional memory (*Halley et al., 2010*). However, it is also possible that the rate at which inositol starvation is perceived might be affected by previous expression of Ino1, leading to a reactivation-specific delay that obscures the effect of memory.

In the presence of inositol, *INO1* transcription is repressed by the combined action of the repressors Opi1 and Ume6 (*Graves and Henry, 2000*; *Jackson and Lopes, 1996*; *White et al., 1991*); loss

of either of these proteins leads to constitutive, high-level expression of *INO1*. Depletion of inositol from the medium slows the rate of phosphatidylinositol biosynthesis, leading to an accumulation of the precursor, phosphatidic acid, which directly binds to and inhibits Opi1 (*Loewen et al., 2004*). Inhibition of Opi1 leads to its dissociation from the promoter, export from the nucleus, and activation of *INO1* (*Brickner and Walter, 2004*; *Loewen et al., 2004*). In cells that have recently expressed Ino1, the extracellular inositol and the inositol produced by Ino1 may exceed that in cells that have not recently expressed Ino1, making them resistant to inositol starvation and delaying the accumulation of phosphatidic acid. To avoid this possible complication, we induced *INO1* transcription by removing Opi1 from the nucleus using the Anchor-Away method (*Haruki et al., 2008*), either in cells that were grown continuously in the presence of inositol (i.e. activation) or in cells that were grown overnight in the absence of inositol and then shifted into medium containing inositol for 3 hr (i.e. reactivation; *Figure 1C*). By removing Opi1 from the nucleus in the presence of inositol, this approach should bypass any insensitivity to inositol starvation, allowing us to directly compare the rate of activation to the rate of reactivation. Upon Anchor Away of Opi1, reactivation was faster than activation, and this effect was lost in both *mrs* and *nup100Δ* mutant cells, confirming that it requires the interaction with the NPC (*Figure 1D*, middle and right panels). Therefore, *INO1* transcriptional memory enhances the rate of transcriptional reactivation.

We next asked if inositol memory promoted competitive fitness by competing pairs of strains with nearly identical plasmids integrated into the genome, differing at a single nucleotide (see Materials and methods). The abundance of each strain was quantified by PCR amplification and Sanger sequencing of a segment encompassing the SNP (either A or C; *Figure 1E*). Mixing strains in various ratios confirmed that this assay is quantitative and accurate over a large dynamic range (*Figure 1F*). Using this assay, we found that, during the initial 3 hr of inositol starvation, cells with inositol memory are more fit than cells experiencing inositol starvation for the first time (*Figure 1G*). Importantly, this fitness benefit is dependent upon Nup100 (*Figure 1G*). Because *INO1* reactivation is not obviously faster than activation under these conditions, this fitness difference may be due to either more uniform expression of *INO1* among cells in the population, similar to other types of transcriptional memory (*Sood and Brickner, 2017*).

Alternatively, fitness may reflect the rate of reactivation of multiple inositol-regulated genes, including *INO1*. Consistent with this idea, we find that *CHO1*, a gene encoding phosphatidyl serine synthase that is also repressed by inositol exhibits all of the hallmarks of transcriptional memory. The *CHO1* locus was tagged with an array of ~128 Lac repressor binding sites in a strain expressing LacI-GFP and the ER (endoplasmic reticulum) membrane marker Pho88-mCherry to localize this gene within a population of live cells (*Robinett et al., 1996*; *Straight et al., 1996*; *D'Urso et al., 2016*; *Egecioglu et al., 2014*). The percentage of cells in which *CHO1* colocalized with Pho88-mCherry at the nuclear periphery was quantified (*Figure 1—figure supplement 1A*). Based on the size of the yeast nucleus, we expect a randomly positioned gene to colocalize with the nuclear envelope in ~27% of cells (blue dashed line; *Brickner et al., 2019*; *Brickner and Walter, 2004*). Following repression, *CHO1* showed Nup100-dependent peripheral localization for at least 3 hr (*Figure 1—figure supplement 1A*). Also, chromatin immunoprecipitation (ChIP) showed both H3K4me2 and RNAPII associated with the *CHO1* promoter 3 hr after repression (*Figure 1—figure supplement 1B*). Finally, *CHO1* showed faster reactivation (*Figure 1—figure supplement 1C*). This protein is required for growth in the absence of inositol (*Atkinson et al., 1980*). Therefore, inositol memory impacts *INO1*, *CHO1*, and potentially other genes, suggesting that the fitness benefit associated with inositol memory is due to the coordinated, enhanced rate of reactivation of a set of genes that promote adaptation to inositol starvation.

Some forms of memory either require transcription during the initial stimulus or reflect the slow dilution of proteins that are expressed in activating conditions (*Kundu and Peterson, 2010*; *Sood and Brickner, 2017*; *Zacharioudakis et al., 2007*). Other forms of memory do not require previous transcription (*Pascual-Garcia et al., 2022*). Therefore, we tested if transcription is required to induce *INO1* memory. First, we determined how long cells need to be starved for inositol to induce *INO1* memory based on its retention at the nuclear periphery. As little as 10 min of inositol starvation resulted in Nup100-dependent peripheral localization 3 hr after shifting back to +inositol (*Figure 1—figure supplement 2A*). This is well before significant transcription has been induced (*Figure 1B*). Using a temperature-sensitive mutation in the large subunit of RNAPII (*rpb1-1*; *Nonet et al., 1987*),

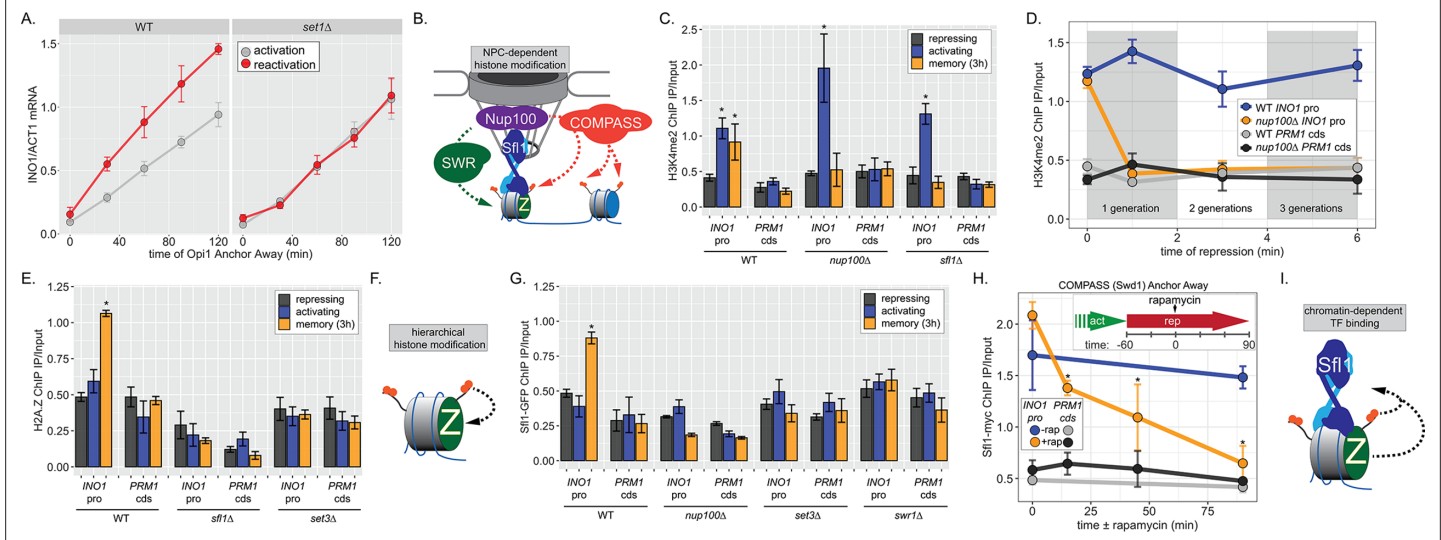

**Figure 2.** *INO1* transcriptional memory requires a positive feedback loop. (**A**) Activation and reactivation of *INO1* in wild type (WT) (left) and *set1Δ* (right) strains upon removal of Opi1 by Anchor Away. Cells were harvested at indicated time points and *INO1* mRNA was quantified relative to *ACT1* mRNA by real time quantitative PCR (RT-qPCR) (*p-value<0.05 from one-tailed t-test comparing reactivation and activation, alternative = greater). Data for the WT strain is the same as *Figure 1D* and is shown for comparison. (**B**) Model for transcription factor (TF)-and NPC-dependent H3K4 dimethylation by COMPASS (orange circles) and SWR (Swi/Snf Related)-dependent H2A.Z incorporation (green Z). For the chromatin immunoprecipitation (ChIP) experiments in panels C, D, E, G, and H: recovery of either the *INO1* promoter or the repressed *PRM1* coding sequence was quantified by RT-qPCR. The average of ≥3 replicates ± SEM is plotted (*p-value<0.05 from one-tailed t-test compared with the repressing condition, alternative = greater). (**C**) ChIP against H3K4me2 in WT, *nup100Δ*, and *sfl1Δ* strains grown under repressing, activating, and memory (3 hr) conditions. (**D**) ChIP against H3K4me2 at the indicated times after switching to repressing conditions in WT and *nup100Δ* strains. The gray and white bars indicate doubling times. (**E**) ChIP against H2A.Z from WT, *sfl1Δ*, or *set3Δ* strains under repressing, activating, or memory (3 hr) conditions. (**F**) Schematic for hierarchical relationship between H2A.Z incorporation and H3K4me2. (**G**) ChIP against Sfl1-GFP in WT, *nup100Δ*, *set3Δ*, and *swr1Δ* strains grown under repressing, activating, or memory (3 hr) conditions. (**H**) ChIP against Sfl1-myc at the indicated times ±1 μg/ml rapamycin in a Swd1 (COMPASS) Anchor Away strain, 1 hr after shifting from activating to repressing conditions. *p-value<0.05 from one-sided t-test compared with the time = 0 min time point, alternative = less. (**I**) Schematic of the requirement for H2A.Z incorporation and H3K4me2 for Sfl1 binding to the *INO1* promoter during memory.

we find that inactivating RNAPII during inositol starvation did not affect retention of *INO1* at the nuclear periphery after repression (*Figure 1—figure supplement 2B*). Finally, to test if transcription is required in cis, we introduced mutations in the TATA box of *INO1*. This mutation blocks *INO1* transcription and leads to an Ino⁻ phenotype (*Figure 1—figure supplement 2C*). However, localization of *INO1* at the nuclear periphery under either activating or memory conditions was unaffected by mutation of the TATA box (*Figure 1—figure supplement 2D*). Therefore, neither transcription of a *trans*-acting factor nor transcription of *INO1* is required for localization at the nuclear periphery during memory. This suggests that *cis*-acting molecular changes associated with the early moments of transcriptional activation induce *INO1* transcriptional memory.

*INO1* memory persists through ~4 cell divisions (6–8 hr; *Brickner et al., 2007*). This duration could reflect either the dilution and degradation of proteins over time or imperfect fidelity of inheritance following DNA replication. To shed light on this, we asked if memory could be extended significantly by arresting/slowing the cell cycle. Indeed, treatment with nocodazole extended *INO1* memory beyond 18 hr (*Figure 1—figure supplement 1*). While dilution is also inhibited by arresting cell division, it seems unlikely that protein production and dilution explain the duration of memory because transcription during inositol starvation is not required. This supports a model in which the duration of memory is regulated by the number of cell divisions or replication cycles.

## A positive feedback loop promotes *INO1* transcriptional memory

To confirm that Opi1 Anchor Away is dependent on the chromatin modifications associated with *INO1* memory, we compared the rate of *INO1* activation and reactivation upon Opi1 Anchor Away in a *set1Δ* strain lacking the catalytic subunit responsible for H3K4 methylation (*Figure 2A*). Indeed, loss

of Set1 had no effect on the rate of *INO1* activation but slowed the rate of reactivation, confirming that H3K4me2 is essential for memory as measured using this system.

To understand the molecular mechanisms controlling the perpetuation and inheritance of *INO1* memory, we tested how the interaction with the NPC impacts chromatin changes and vice versa (*Figure 2B*). ChIP in wild type (WT) cells reveals that H3K4me2 is observed under activating and memory conditions (*Figure 2C*), while H2A.Z upstream of the TSS is only observed during memory (*Figure 2C*; *Light et al., 2010*). The Sfl1 TF binds to the *INO1* promoter specifically during memory, requires the MRS DNA zip code, and is both necessary and sufficient to induce Nup100-dependent peripheral localization (*D'Urso et al., 2016*). In strains lacking either Nup100 or Sfl1, H3K4me2 is lost during memory (*Figure 2C*) and the rate of reactivation is slowed (*Figure 1D*; *D'Urso et al., 2016*; *Light et al., 2010*). Furthermore, H3K4me2 is rapidly lost in the *nup100Δ* strain upon shifting from activating to repressing conditions (*Figure 2D*). Likewise, incorporation of H2A.Z during memory also requires Sfl1 (*Figure 2E*), as has been seen for Nup100 (*Light et al., 2010*). Thus, the interaction with the NPC stimulates both H3K4me2 and H2A.Z incorporation during *INO1* memory (*Figure 2E*, left).

To explore how chromatin modifications impact each other, we performed ChIP against H2A.Z in a strain lacking Set3, a structural subunit of the SET3C histone deacetylase that binds H3K4me2 and is essential for maintaining H3K4me2 during memory (*D'Urso et al., 2016*; *Kim and Buratowski, 2009*; *Light et al., 2013*). Strains lacking Set3 also failed to incorporate H2A.Z during *INO1* memory, suggesting that H2A.Z incorporation requires Sfl1/Nup100 and H3K4me2. Because H3K4me2 during memory does not require H2A.Z (*Light et al., 2013*), this suggests a hierarchical relationship between dimethylation of H3K4 and incorporation of H2A.Z (*Figure 2F*).

Finally, we asked if Sfl1 binding to the *INO1* promoter during memory requires either interaction with the NPC or chromatin modifications. Surprisingly, loss of Nup100, Set3, or Swr1 (the catalytic subunit of the SWR complex, which incorporates H2A.Z into chromatin; *Mizuguchi et al., 2004*) disrupted binding of Sfl1 to the *INO1* promoter during memory (*Figure 2G*). In other words, while Sfl1 is required for interaction of the *INO1* gene with the NPC and chromatin modifications, Nup100 and chromatin modification are also required for Sfl1 binding to the *INO1* promoter. We confirmed this dependence on H3K4me2 by conditional inactivation of COMPASS (the histone methyltransferase) using Anchor Away of Swd1 after establishing memory (*D'Urso et al., 2016*). Removing COMPASS from the nucleus leads to loss of H3K4me2 over the *INO1* promoter within ~60 min (*D'Urso et al., 2016*). In cells that have established memory, Sfl1 was bound to the *INO1* promoter (*Figure 2H*, t=0), but upon Anchor Away of COMPASS, Sfl1 binding was lost rapidly (*Figure 2H*). Thus, Sfl1 binding to the *INO1* promoter during memory requires H3K4me2 (*Figure 2I*). Together, these data suggest that *INO1* memory involves positive feedback between Sfl1-dependent interaction with the NPC and NPC-dependent H3K4me2 and H2A.Z incorporation.

## Two Heat Shock Factor-related transcription factors are required for *INO1* transcriptional memory

The MRS DNA zip code (5'-TCCTTCTTTCCC-3'; *Light et al., 2010*) contains sequences reminiscent of the trinucleotide repeats within heat shock elements (5'-TTC-3'). This aided in the identification of Sfl1, which possesses an Hsf1-like DNA binding domain (*D'Urso et al., 2016*). In budding yeast, there are three other TFs with similar DNA binding domains (Hms2, Mga1, and Skn7), which we also tested for a role in *INO1* memory (*Figure 3—figure supplement 1*). Of these proteins, only Hms2 was required for localization of *INO1* at the nuclear periphery during memory (*Figure 3—figure supplement 1*). Loss of Mga1 resulted in constitutive *INO1* localization at the periphery, while loss of Skn7 had no effect (*Figure 3—figure supplement 1*). This suggests that Sfl1 and Hms2 are specifically required for *INO1* transcriptional memory and that Mga1 may play a role in negatively regulating peripheral localization.

If Hms2 were required for *INO1* memory, we expected that it would bind to the *INO1* promoter during memory in an MRS-dependent manner and be required for the molecular outputs of memory. ChIP against Hms2-myc revealed that it bound to the *INO1* promoter both in activating and memory conditions and that binding was lost in the *mrs* mutant (*Figure 3A*). This suggests that Hms2 binds to the MRS both before and after establishing memory. Under activating conditions, loss of Hms2 did not affect RNAPII binding or methylation of H3K4, but during memory, loss of Hms2 led to loss of both (*Figure 3B*). Furthermore, loss of Hms2 led to a specific decrease in the rate of reactivation

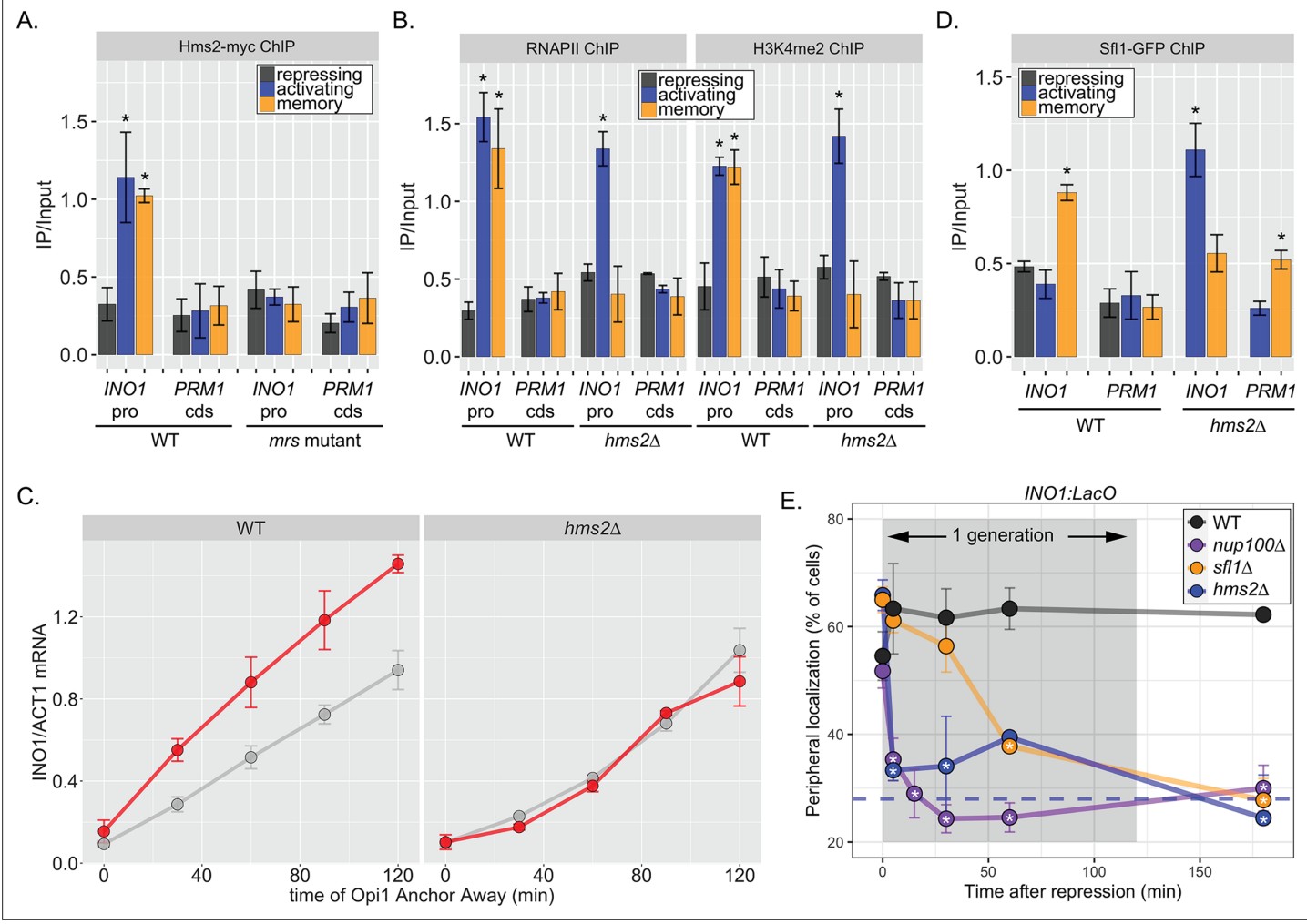

**Figure 3.** Two different Hsf1-like TFs are required for inositol memory. (**A–C**) Chromatin immunoprecipitation (ChIP) against Hms2-myc (A), RNAPII (B, left), H3K4me2 (B, right), or Sfl1-GFP (**D**) in the indicated strains grown under activating, repressing, or memory (3 hr) conditions. Recovery of either the *INO1* promoter or the repressed *PRM1* coding sequence was quantified by real time quantitative PCR (RT-qPCR) and the average of ≥3 replicates ± SEM is plotted (*p-value<0.05 from one-tailed t-test compared with the repressing condition, alternative = greater). (**C**) Activation and reactivation of *INO1* in wild type (WT) (left) and *hms2Δ* (right) strains upon removal of Opi1 by Anchor Away. Cells were harvested at indicated time points and *INO1* mRNA was quantified relative to *ACT1* mRNA by RT-qPCR (*p-value<0.05 from one-tailed t-test comparing reactivation and activation, alternative = greater). Data from the WT strain is the same as *Figure 1D* and is shown for comparison. (**E**) Peripheral localization of *INO1* in either WT, *sfl1Δ*, *hms2Δ*, or *nup100Δ* strains. At t=0, inositol was added to cells growing without inositol and peripheral localization was scored at the indicated times. The doubling time of this strain (~120 min) is indicated. The average of three biological replicates ± SEM is plotted and each biological replicate ≥30 cells. Blue hatched line: expected peripheral localization for a randomly localized gene. *p-value<0.05 from one-tailed t-test compared with time = 0, alternative = less.

The online version of this article includes the following figure supplement(s) for figure 3:

**Figure supplement 1.** Some, but not all, Hsf1-like TFs are required for *INO1* transcriptional memory.

upon Opi1 Anchor Away (*Figure 3C*). Therefore, Hms2 binds during both activation and memory but is specifically required for *INO1* memory.

Because Hms2 bound prior to Sfl1, we tested if Hms2 impacts Sfl1 binding. ChIP against Sfl1-GFP revealed that, in cells lacking Hms2, Sfl1 bound during activating conditions instead of memory (*Figure 3D*). Furthermore, loss of either TF or Nup100 led to rapid loss of *INO1* localization at the nuclear periphery, albeit with slightly different kinetics (*Figure 3E*). All of the mutants resulted in a loss of localization within one generation (denoted by gray box), but loss of Hms2 and Nup100 had the most immediate effect. Thus, our genetic results suggest that Hms2 prevents Sfl1 binding to the active *INO1* promoter but is required for Sfl1 binding during memory, perhaps binding as a heterodimer.

## The binding of RNAPII during memory depends on H3K4me2, but H3K4me2 does not depend on RNAPII binding

One of the hallmarks of *INO1* memory is the recruitment of poised RNAPII PIC lacking the Cdk7 kinase (Kin28 in budding yeast) but including the Cdk8 Mediator kinase (Ssn3 in budding yeast; *D'Urso et al., 2016*). Anchor Away of Ssn3 from the nucleus disrupts poised RNAPII from the *INO1* promoter during memory, suggesting that Cdk8 is required for recruitment or maintenance of this poised PIC (*D'Urso et al., 2016*). To test if the kinase activity is required for RNAPII poising during *INO1* memory, we constructed an analog-sensitive allele of Ssn3 (*ssn3-as*; Y236G) that is inhibited by the ATP analog 1-Napthyl-PP1 (NaPP1; *Bishop et al., 2000*; *Liu et al., 2004*). Inhibition of Ssn3 resulted in loss of RNAPII from the *INO1* promoter during memory, without affecting the recruitment of RNAPII under activating conditions (*Figure 4A*). In contrast, Anchor Away of the budding yeast TATA binding protein (Spt15) from the nucleus disrupted RNAPII binding to the *INO1* promoter under both activating and memory conditions (*Figure 4A*). Inhibition of Ssn3 specifically slowed the rate of reactivation of *INO1* (*Figure 4B*) and eliminated the fitness benefit of memory (*Figure 4C*), confirming that poised RNAPII is important to increase the rate of reactivation.

H3K4 methylation is stimulated by active RNAPII (*Bae et al., 2020*; *Krogan et al., 2003*; *Krogan et al., 2002*). Consistent with this, Anchor Away of Spt15 results in loss of H3K4me2 from the active *INO1* promoter (*Figure 4D*). However, neither depletion of Spt15 nor inhibiting Ssn3 affected H3K4 dimethylation under memory conditions (*Figure 4C*). This suggests that poised RNAPII at the *INO1* promoter during memory is not required for dimethylation of H3K4. To confirm this idea, we performed ChIP against H3K4me2 and H3K4me3 under repressing and memory conditions in either WT or *ino1-tataΔ* mutant strains. Mutation of the TATA box from the *INO1* promoter, while blocking *INO1* transcription (*Figure 1—figure supplement 1*), does not affect the H3K4 dimethylation during memory (*Figure 4E*). Thus, RNAPII recruitment is not essential for H3K4 dimethylation observed during memory.

The inhibition of analog sensitive kinases by NaPP1 is readily reversible (*Wan et al., 2006*). Therefore, we asked if the loss of poised RNAPII from the *INO1* promoter upon inhibition of Ssn3 during memory was reversible upon removal of NaPP1 (*Figure 4G*). Cells were shifted from activating to memory conditions for 1 hr, Ssn3 was inhibited for 30 min to disrupt RNAPII binding, and then NaPP1 was washed away (*Figure 4F*, top). Inhibiting Ssn3 led to loss of RNAPII association with the *INO1* promoter within 30 min (*Figure 4H*, left). Upon removing NaPP1, RNAPII binding was recovered fully within 90 min (*Figure 4H*, left). Therefore, RNAPII is not required to maintain the *INO1* promoter in a memory-compatible state.

To ask if chromatin is critical for the return of poised RNAPII, we asked if RNAPII could return to the *INO1* promoter following inhibition of Ssn3 in the absence of H3K4me2 (*Figure 4G*, bottom). H3K4me2 was removed by auxin-inducible degradation of Set3 (Set3-AID; *Nishimura et al., 2009*), which is required to maintain H3K4me2 over the *INO1* promoter during memory (*D'Urso et al., 2016*). In the *ssn3-as SET3-AID* strain, NaPP1 alone had no effect on H3K4me2 over the *INO1* promoter during memory, while NaPP1 plus auxin resulted in rapid loss (*Figure 4F*). In such cells lacking H3K4me2, RNAPII failed to return to the *INO1* promoter after washing away NaPP1 (*Figure 4G*, bottom and *Figure 4H*, right). We conclude that dimethylation of H3K4 is necessary for recruitment of poised RNAPII to the *INO1* promoter during memory.

## The Paf1 complex (Paf1C) subunit Leo1 is specifically required for memory

The work on *INO1* suggests that the molecular requirements for dimethylation of H3K4 at active promoters are different from the molecular requirements for dimethylation of H3K4 at the same promoters during memory; while active promoters require RNAPII for H3K4 methylation, poised promoters do not. Therefore, we asked if the Paf1C, a conserved factor required for H3K4 methylation, plays a role in H3K4 dimethylation during memory. The yeast Paf1C (Cdc73, Ctr9, Leo1, Paf1, and Rtf1) associates with active RNAPII and promotes methylation of H3K4 by recruiting COMPASS (*Krogan et al., 2003*). Loss of certain Paf1C proteins (Paf1, Ctr9, and Rtf1) completely blocks H3K4 methylation, while loss of Cdc73 and Leo1 have no obvious effect (*Krogan et al., 2003*; *Ng et al., 2003a*; *Ng et al., 2003b*). We assessed how loss of these factors affected H3K4me2 and H3K4me3 over the *INO1* promoter under repressing, activating, and memory conditions. As expected, loss of

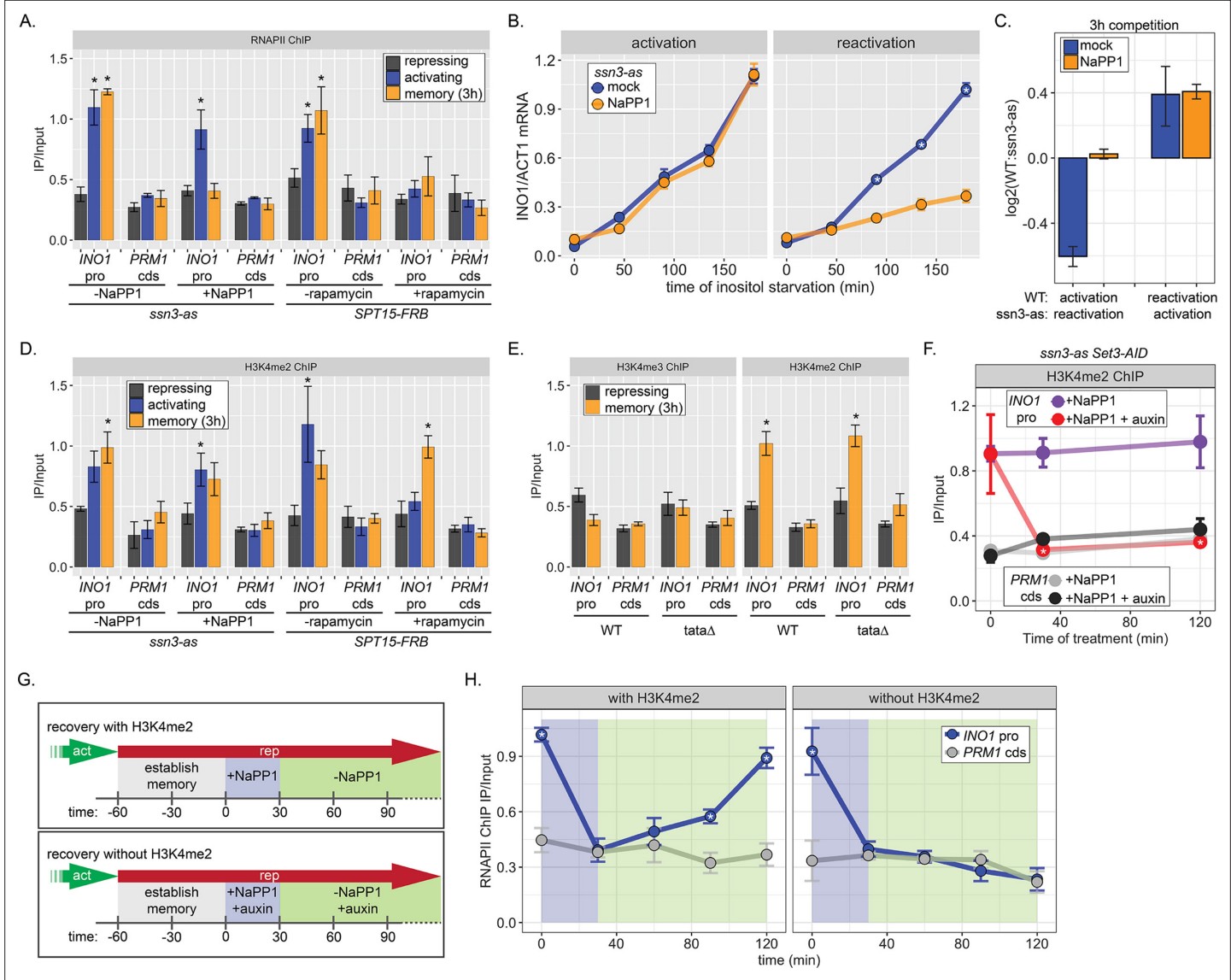

**Figure 4.** RNA polymerase II (RNAPII) binding during *INO1* memory requires H3K4me2, but H3K4 dimethylation does not depend on RNAPII. Chromatin immunoprecipitation (ChIP) against RNAPII (**A**) or H3K4me2 (**D**) in *ssn3* analog-sensitive (*ssn3-as*) and *SPT15-FRB* Anchor Away strains upon addition of either 1-Napthyl-PP1 (NaPP1) or rapamycin for 1 hr as indicated. Cells were grown in repressing, activating, or memory (3 hr) conditions. *p<0.05 from one-tailed t-test compared with repressing condition, alternative = greater. (**B**) Activation (left) and reactivation (right) of *INO1* in the *ssn3-as* strain treated either with DMSO (dimethyl sulfoxide; mock) or NaPP1 for upon inositol starvation (1 hr pretreatment or treatment). Cells were harvested at indicated time points and the *INO1* mRNA was quantified relative to *ACT1* mRNA by real time quantitative PCR. *p<0.05 from one-tailed t-test comparing between mock and NaPP1, alternative = greater. (**C**) The log₂ ratio of the indicated strains to each other after competition for 3 hr in the absence of inositol. (**E**) ChIP against H3K4me3 (left) or H3K4me2 (right) in wild type (WT) and *ino1-tata* strains. Cells were grown in repressing (+inositol) or memory conditions (+inositol → −inositol, 1 hr →+inositol, 3 hr). *p<0.05 from one-tailed t-test compared with repressing condition, alternative = greater. (**F**) ChIP against H3K4me2 after establishing memory for 1 hr, followed by addition of either NaPP1 or NaPP1 and auxin at t=0. * p-value<0.05 from one-tailed t-test comparing NaPP1 treated and NaPP1+auxin treated samples at each time, alternative = less. (**G**) Schematic for experiment in (H) to monitor RNAPII recruitment with (top) or without (bottom) H3K4me2. NaPP1 was added with or without 0.5 mM auxin for 30 min before removing NaPP1. (**H**) ChIP against RNAPII, following the experimental set up in (G) with cells crosslinked at the indicated times. For panels A, D, E, F, and H, recovery of the *INO1* promoter or the *PRM1* coding sequence (negative control locus) was quantified relative to input by qPCR the averages of three biological replicates ± SEM were plotted; *p-value<0.05 from one-tailed t-test comparing ChIP of *INO1* promoter to *PRM1* cds at each time, alternative = greater.

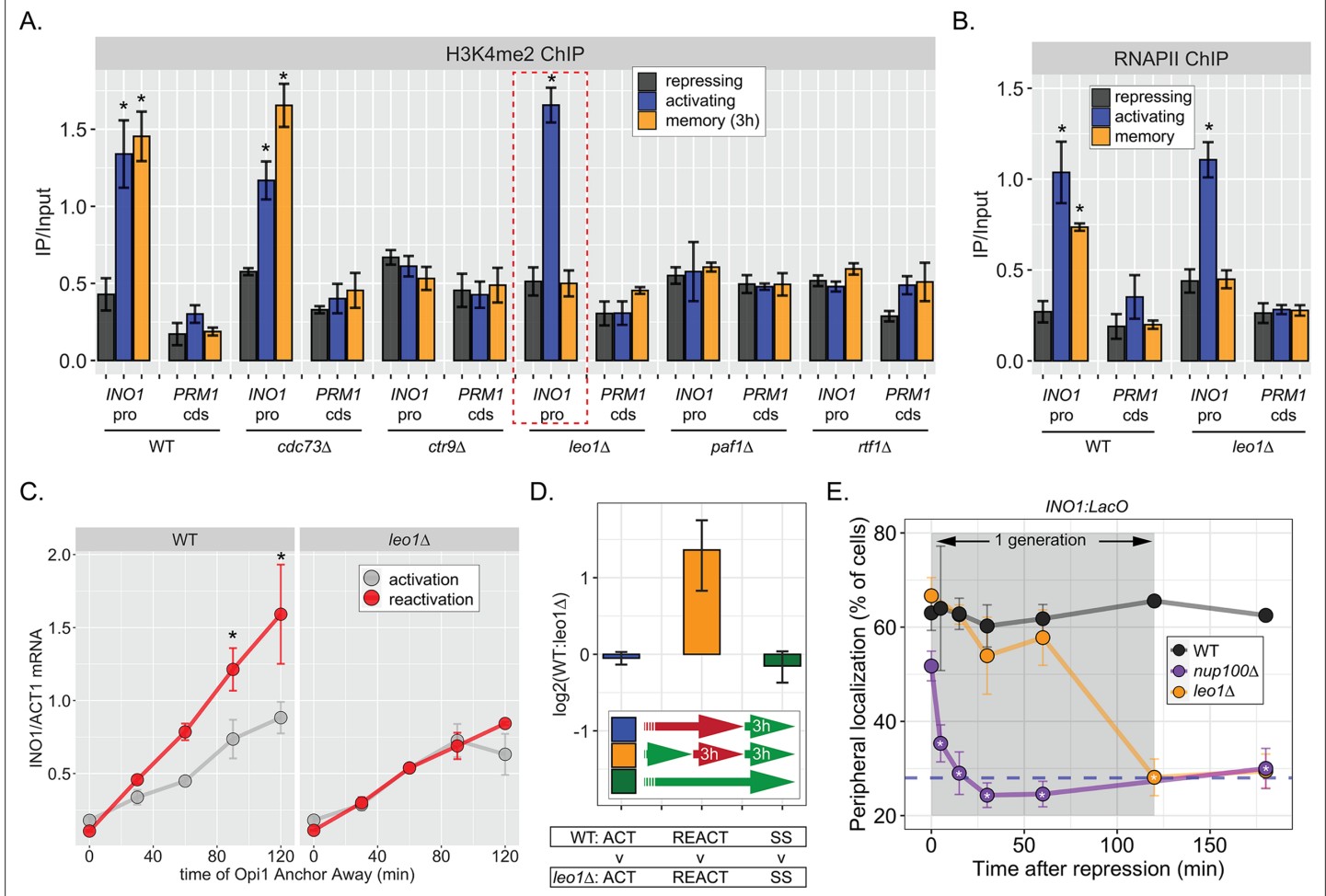

**Figure 5.** The Leo1 protein of the Paf1 complex is specifically required for *INO1* memory. (**A**) Chromatin immunoprecipitation (ChIP) against H3K4me2 in the indicated strains under repressing (+inositol), activating (−inositol), or memory (−inositol →+inositol, 3 hr) conditions, highlighting the effects of the *leo1Δ* mutation (dashed red box). (**B**) ChIP against RNA polymerase II in wildtype (WT) and *leo1Δ* strains repressing, activating (−inositol), or memory (−inositol →+inositol, 3 hr) conditions. For A & B: recovery of *INO1* promoter or *PRM1* coding sequence (negative control) were quantified by quantitative PCR and the averages of three biological replicates ± SEM are plotted. For A and B, *p<0.05 from one-tailed t-test comparing against repressed condition, alternative = greater. (**C**) Activation and reactivation of *INO1* in WT (left) or *leo1Δ* (right) strains upon Anchor Away of Opi1. Cells were harvested at indicated times and *INO1* mRNA was quantified relative to *ACT1* mRNA by real time quantitative PCR. *p<0.05 from one-tailed t-test comparing activation to reactivation, alternative = greater. (**D**) Competitive fitness of WT vs *leo1Δ* strains competed for 3 hr in the absence of inositol during activation (ACT,+inositol → −inositol, blue), reactivation (REACT, −inositol →+inositol (3 hr) → −inositol, orange), or steady state (SS, −inositol → −inositol, green). The ratio of the relative abundance of the WT:*leo1Δ* strains was quantified and expressed as a log2 ratio ± SEM. (**E**) Peripheral localization of *INO1* in WT, *nup100Δ* (replotted from *Figure 3*), and *leo1Δ* strains shifted from activating (−inositol) to repressing (memory) conditions for the indicated times. The gray box indicates the approximate doubling time of this strain. The average of ≥3 biological replicates ± SEM is plotted and each biological replicate ≥30 cells.

The online version of this article includes the following figure supplement(s) for figure 5:

**Figure supplement 1.** Effects of loss of Leo1 on H3K4me3 and *INO1* localization.

Ctr9, Paf1, or Rtf1 blocked all methylation and loss of Cdc73 had no effect (*Figure 5A*, *Figure 5—figure supplement 1A*). However, strains lacking Leo1 showed normal levels of H3K4me3 and H3K4me2 over the active *INO1* promoter, but no H3K4 methylation over the *INO1* promoter during memory (*Figure 5A*, *Figure 5—figure supplement 1*). Thus, Paf1C is required for H3K4 dimethylation during memory and the Leo1 subunit plays a memory-specific role in promoting H3K4me2.

Consistent with an essential and specific role in *INO1* memory, loss of Leo1 also disrupted *INO1* localization at the nuclear periphery (*Figure 5—figure supplement 1B*) and RNAPII binding to the *INO1* promoter (*Figure 5B*) during memory, slowed the rate of transcriptional reactivation (*Figure 5C*),

and erased the fitness benefit associated with *INO1* transcriptional memory (***Figure 5D***). This essential and specific role of Leo1 in promoting *INO1* memory supports the idea that the molecular mechanism of H3K4 methylation during memory is carried out by an overlapping, but distinct, pathway from that responsible for H3K4 methylation during transcription.

When we examined the rate at which *leo1Δ* lost memory, we found that *INO1* remained at the nuclear periphery for ~60 min before repositioning to the nucleoplasm rapidly (***Figure 5E***). In contrast, loss of Nup100 (or the mutations in the MRS; ***Light et al., 2010***), resulted in immediate loss

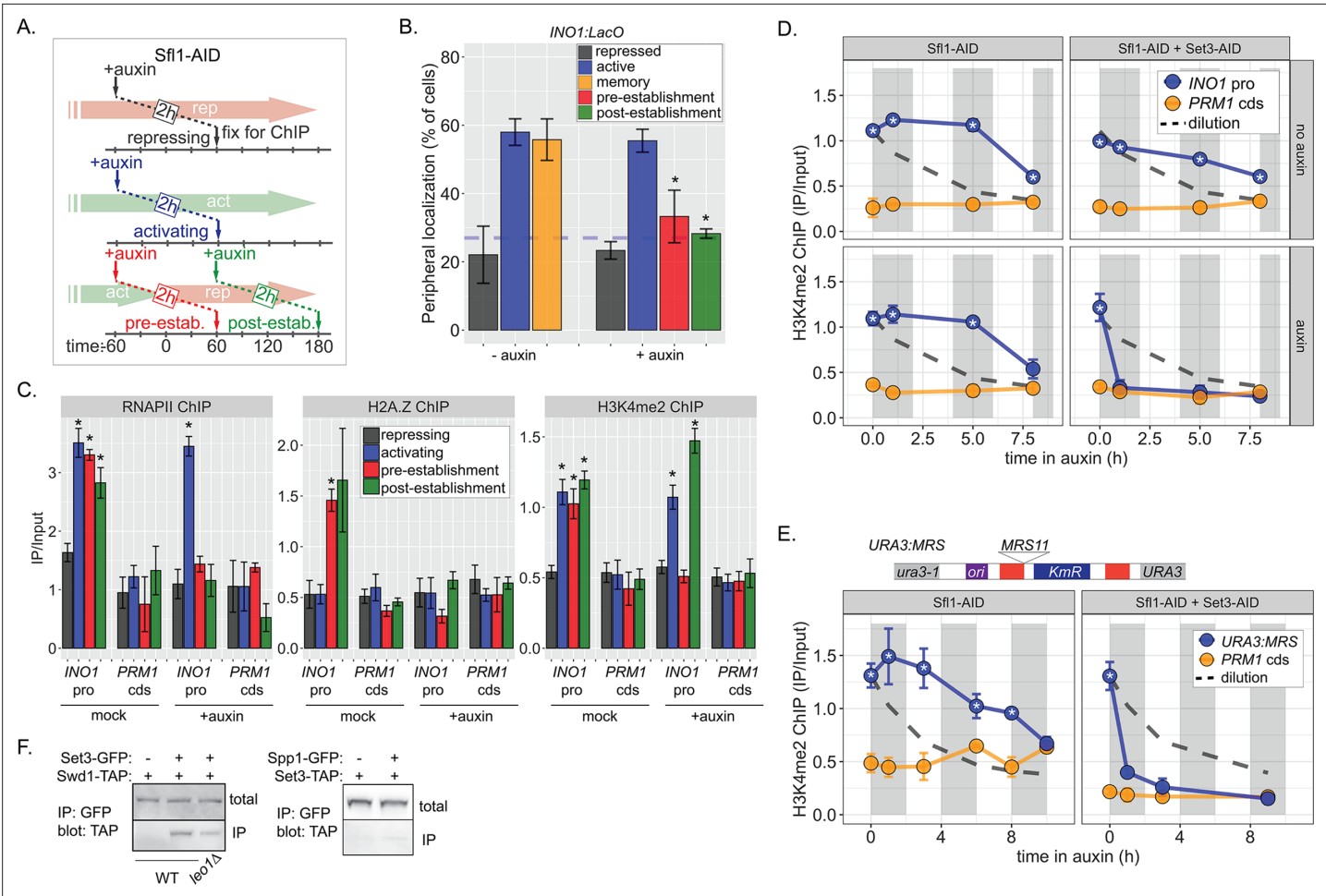

**Figure 6.** Distinct molecular requirements for establishment and inheritance of H3K4me2 during *INO1* memory. (**A**) Experimental set-up to test the role of Sfl1 in establishment and inheritance of *INO1* memory, using auxin-inducible degradation of Sfl1 before or after establishing memory. The top arrow indicates when auxin was added; the bottom arrow indicates when cells were fixed for chromatin immunoprecipitation (ChIP). Using this approach, peripheral localization of *INO1* (**B**) or ChIP against RNAPII (**C**, left), H2A.Z (**C**, middle), or H3K4me2 (**C**, right) was measured under the indicated conditions ±0.5 mM auxin. Peripheral localization is the average of three biological replicates ± SEM; each biological replicate ≥30 cells. Blue hatched line: expected peripheral localization for a randomly localized gene. *p<0.05 from one-tailed t-test comparing auxin treated to untreated, alternative = less. (**D**) ChIP against H3K4me2 in *Sfl1-AID* (left) and *Sfl1-AID+Set3* AID (right) strains either without auxin (top) or after addition of auxin (lower). Auxin was added after 1 hr of repression. For panels D and E, vertical gray and white bars represent estimated generation times and dashed line represents the expectation from perfect retention of H3K4me2, followed by dilution through DNA replication (i.e. $t_{1/2}$ = the doubling time). (**E**) Schematic of insertion of 11 bp MRS at the *URA3* locus in the *URA3:MRS* strain (top) and ChIP against H3K4me2 in *URA3:MRS Sfl1-AID* (left) and *URA3:MRS Sfl1-AID+Set3* AID (right) strains after addition of auxin (lower). Panels C, D, and E: recovery of *INO1* promoter or *PRM1* coding sequence or *URA3:MRS* region was quantified by quantitative PCR relative to input and are the averages of three biological replicates ± SEM. *p<0.05 from one-tailed t-test comparing to recovery of each DNA in the repressed condition (C) or to *PRM1* cds (D & E), alternative = greater. (**F**) Co-immunoprecipitation of Set3-GFP and Swd1-TAP (left), Spp1-GFP and Set3-TAP (right) from the indicated strains. The Green fluorescent protein (GFP)-tagged proteins were immunoprecipitated with anti-GFP nanobodies; recovery of Swd1-TAP and Set3-TAP were monitored by immunoblotting with anti-TAP antibody.

The online version of this article includes the following figure supplement(s) for figure 6:

**Figure supplement 1.** Rapid loss of H3K4me2 from the *INO1* promoter in the absence of transcriptional memory.

of peripheral localization upon repression (*Figure 5E*). Thus, Leo1 is not required to establish memory, but is required to either sustain or inherit memory.

## RNAPII-independent H3K4 dimethylation is mitotically heritable

Because factors such as Leo1 are involved in sustaining, but not establishing memory, we asked Sfl1 is involved in the establishment of memory, its inheritance, or both. We inactivated Sfl1 using auxin-induced degradation either before or after establishing memory (*Figure 6A*). Degrading Sfl1-AID either before or after establishing memory led to loss of peripheral localization (*Figure 6B*), confirming that Sfl1 is critical for interaction with the NPC and that the auxin-induced degradation was complete. Likewise, degrading Sfl1 either before or after establishing memory led to loss of H2A.Z and RNAPII during memory (*Figure 1C*, left and middle panels). However, H3K4me2 behaved differently. Degrading Sfl1 prior to establishing memory prevented H3K4me2 during memory (*Figure 1C*, right panel), confirming that Sfl1-AID leads to loss of Sfl1 function. However, in cells in which Sfl1-AID was degraded after establishing memory, H3K4me2 persisted (*Figure 1C*, right panel). Therefore, once memory has been established, H3K4 dimethylation can persist for up to 2 hr in the absence of Sfl1.

The observation in *Figure 1C* raised the possibility that RNAPII-independent H3K4me2 is stable or heritable in the absence of factors required for its initial deposition. However, to establish the framework for testing this possibility, we first assessed the stability of RNAPII-dependent H3K4 methylation. To do this, we followed H3K4me2 by ChIP over the *mrs* mutant *INO1* promoter following repression, in the absence of transcriptional memory. For comparison, we projected the amount of this mark that would remain if it were neither removed nor deposited (*Figure 6—figure supplement 1A*, dashed line, representing dilution through DNA replication with $t_{1/2}$=120 min). H3K4me2 was lost rapidly over the *mrs* mutant *INO1* promoter upon repression (*Figure 6—figure supplement 1A*), confirming that RNAPII-dependent H3K4 dimethylation is neither stable nor heritable. The active removal of H3K4 methylation upon repression is likely mediated by the H3K4-specific demethylase, Jhd2 (*Liang et al., 2007*).

ChIP against H3K4me2 following degradation of Sfl1 (post-establishment; *Figure 6A*) revealed that the *INO1* promoter remained associated with this mark for 5–8 hr longer (3–4 generations; *Figure 1D*, left panels). The rate of loss was unaffected by removal of Sfl1 (*Figure 1D*, left panels), indicating that this TF is not required for inheritance of H3K4me2 during memory. In contrast, in strains in which both Sfl1-AID and Set3-AID were degraded, H3K4me2 was lost rapidly (*Figure 6D*, right panels). Therefore, once established, Sfl1-/Nup100-dependent H3K4me2 is actively reincorporated and efficiently inherited after DNA replication and this requires Set3.

To explore how long H3K4me2 can be inherited, we exploited the MRS zip code. Introduction of a single copy of this 11-base pair element near the *URA3* locus (*Figure 6E*, top) recapitulates important aspects of transcriptional memory: peripheral localization, H3K4me2, and H2A.Z incorporation, but not RNAPII binding (*Light et al., 2013*; *Light et al., 2010*). We have interpreted this to suggest that interaction of the NPC is sufficient to induce the chromatin changes associated with memory, but that recruitment of RNAPII requires *cis* acting promoter elements (*D'Urso et al., 2016*; *Light et al., 2013*; *Light et al., 2010*). Importantly, the changes induced by the MRS are constitutive, allowing us to assess the duration of heritability without considering the normal mechanisms that regulate the duration of *INO1* memory. Following Sfl1 degradation, *URA3:MRS* localization to the nuclear periphery was lost (*Figure 6—figure supplement 1B*), but H3K4me2 was maintained for ≥8 hr (~4 generations; doubling time indicated by the gray and white bars; *Figure 6E*, left). This vastly exceeds either the persistence expected from simple dilution of this mark through DNA replication (dashed line) or that observed for RNAPII-dependent H3K4me2 (*Figure 6—figure supplement 1*). Degradation of both Sfl1 and Set3 led to rapid loss of H3K4me2, supporting the requirement for Set3 in this inheritance (*Figure 6E*, right). Therefore, RNAPII-independent H3K4 dimethylation is actively reincorporated to allow mitotic inheritance.

The epigenetic maintenance and spreading of histone marks generally requires that the enzyme that catalyzes the deposition of the mark (the writer) physically interact with a protein/complex that recognizes the mark (the reader; *Francis, 2009*; *Ragunathan et al., 2015*). Set3 physically interacts with H3K4me2 through its plant homeodomain (PHD) finger and replacing tryptophan 140 with alanine disrupts this interaction (*Kim and Buratowski, 2009*). This mutation also disrupts H3K4me2

and RNAPII binding to the *INO1* promoter during memory (*D'Urso et al., 2016*), suggesting that it may function as a reader of this mark, protecting it from removal. We hypothesized that it may also serve to recruit the writer of the mark by physically interacting with COMPASS. Indeed, co-immunoprecipitation of Set3-GFP recovered the COMPASS subunit Swd1-TAP and this interaction was reduced in a strain lacking Leo1 (*Figure 6F*, left). This interaction was not observed when the Spp1 subunit of COMPASS was tagged with GFP (*Figure 6F*, middle), consistent with SET3C interacting with the memory-specific form of COMPASS, which lacks Spp1 (*D'Urso et al., 2016*). Therefore, the reader and writer of H3K4 dimethylation physically interact and this interaction is stimulated by Leo1.

## Discussion

Epigenetic phenomena result in heritable changes in phenotype without changes in DNA sequence (*Nanney, 1958*; *Waddington, 2012*). The classic examples are stable states, metastable states, or regulated switches in state. Stable states include transcriptionally silent subtelomeric, pericentromeric regions, or mating type loci (*Gartenberg and Smith, 2016*; *Holoch and Moazed, 2015*; *Zofall and Grewal, 2006*). Metastable states include colony morphology switching in microbes (*Klar et al., 2001*; *Slutsky et al., 1987*) and position effect variegation in animals (*Elgin and Reuter, 2013*). Stable switching of transcriptional states occurs during development, such as silencing of the mammalian X chromosome or establishment of the gene expression programs required for differentiation (*Campos et al., 2014*; *Galupa and Heard, 2018*; *Gibney and Nolan, 2010*; *Tee and Reinberg, 2014*). Such phenomena often require histone modifications, sometimes called epigenetic marks. However, histone modifications are generally regulated by sequence-specific DNA binding proteins that interact with *cis*-acting, genetically encoded DNA elements (*Holoch et al., 2021*; *Holoch and Moazed, 2015*; *Laprell et al., 2017*). Thus, signal transduction that alters the activity of a TF can produce a heritable change in phenotype that requires histone modifications without the histone modification itself being heritable. Unlike DNA methylation, in which the methyl marks retained on one strand after replication serve to stimulate methylation of the other (*Edwards et al., 2017*; *Hermann et al., 2004*), histone/nucleosome modifications are not necessarily re-incorporated at the same location after DNA replication (*Escobar et al., 2021*; *Wang et al., 2009*). However, both H3K9 methylation (constitutive heterochromatin; *Ragunathan et al., 2015*; *Zhang et al., 2008*) and H3K27 methylation (facultative heterochromatin; *Coleman and Struhl, 2017*; *Hansen and Helin, 2009*; *Hansen et al., 2008*; *Margueron et al., 2009*) can be inherited through mitosis following removal of the initiating factors (albeit not as stably as with those factors). Furthermore, elegant proximity labeling studies show that nucleosomes at certain silent loci are re-incorporated locally after DNA replication (*Escobar et al., 2019*). Thus, mitotic inheritance requires both the local reincorporation of marked nucleosomes and reinforcement of these modifications through covalent modification of unmarked nucleosomes following DNA replication. The histone marks for which there is the clearest data for heritability share several features: (1) they tend to occur over large regions, encompassing many nucleosomes, (2) they are associated with transcriptional repression, and (3) inheritance requires an interaction between a writer and a reader that recruits or stimulates the writer.

Here we find that H3K4 dimethylation of nucleosomes over the same location in the genome can either be stable and heritable or unstable and rapidly removed, depending on the mechanism by which it is deposited. During active transcription, nucleosomes in the *INO1* promoter are marked with histone acetylation (*D'Urso et al., 2016*; *Rundlett et al., 1998*), H3K4me3, and H3K4me2 (*Santos-Rosa et al., 2002*). This H3K4 methylation is catalyzed by COMPASS and requires RNAPII and the preinitiation complex (*Figure 4C*; *D'Urso et al., 2016*). In mutants that lack memory, these marks are rapidly removed upon transcriptional repression (*Figure 6—figure supplement 1*; *D'Urso et al., 2016*). In contrast, during memory, histone acetylation and H3K4me3 are lost and H3K4me2 is deposited (*Figure 6C*; *D'Urso et al., 2016*). This mechanism of H3K4 dimethylation during memory is mechanistically distinct from that observed during transcription: it is catalyzed by Spp1⁻ COMPASS, does not require RNAPII and requires SET3C, Leo1, Sfl1, and Nup100 (*D'Urso et al., 2016*; *Light et al., 2013*). Critically, once established, H3K4me2 is both stable and inherited for ~4 cell divisions following degradation of Sfl1 (*Figure 6*). This is distinct from the effects of inactivating COMPASS or SET3C (*Figures 4 and 6*; *D'Urso et al., 2016*). Therefore, H3K4me2 can be maintained and inherited in the absence of an essential initiating factor but continuously requires the writer and reader. We

conclude that, as with histone marks associated with heterochromatin (*Reinberg and Vales, 2018*), a histone mark associated with transcriptional poising is epigenetically inherited.

How is heritable H3K4me2 distinguished from unstable H3K4me2? It is possible that the heritable signal comprises H3K4me2 and additional histone marks. Although the incorporation of H2A.Z during memory is an intriguing possibility for a second signal, it is not essential for H3K4 dimethylation (*Light et al., 2013*). Alternatively, unacetylated histones may stimulate heritable H3K4 dimethylation, perhaps by regulating the activity of Spp1⁻ COMPASS or the Paf complex via Leo1. This may explain why the SET3C histone deacetylase is the reader for this mark.

Whereas H3K9 methylation or H3K27 methylation is generally very stable and state switching promotes long-term or permanent switches, transcriptional memory persists for shorter timescales, generally between 4 and 14 mitotic divisions. This limited inheritance may reflect a balance between the fitness benefits of memory and its costs in a fluctuating environment. Regardless, our current understanding suggests that the duration of memory relates to limits on inheritance following each round of DNA replication. Whereas H3K9 methylation or H3K27 methylation generally covers tens to hundreds of thousands of base pairs (*Barski et al., 2007*; *Cutter DiPiazza et al., 2021*; *Pauler et al., 2009*; *Yu et al., 2014*), the chromatin changes associated with epigenetic transcriptional memory are more local, covering hundreds of base pairs (*D'Urso et al., 2016*). Following DNA replication, nucleosomes are randomly segregated into the two daughter molecules (*Petryk et al., 2018*; *Yu et al., 2018*). Approximately half of the nucleosomes should retain regulatory marks and the inheritance of those marks requires recognizing these nucleosomes and modifying adjacent, unmarked nucleosomes (*Escobar et al., 2021*; *Loyola et al., 2006*). If recognition and/or modification is imperfect, the

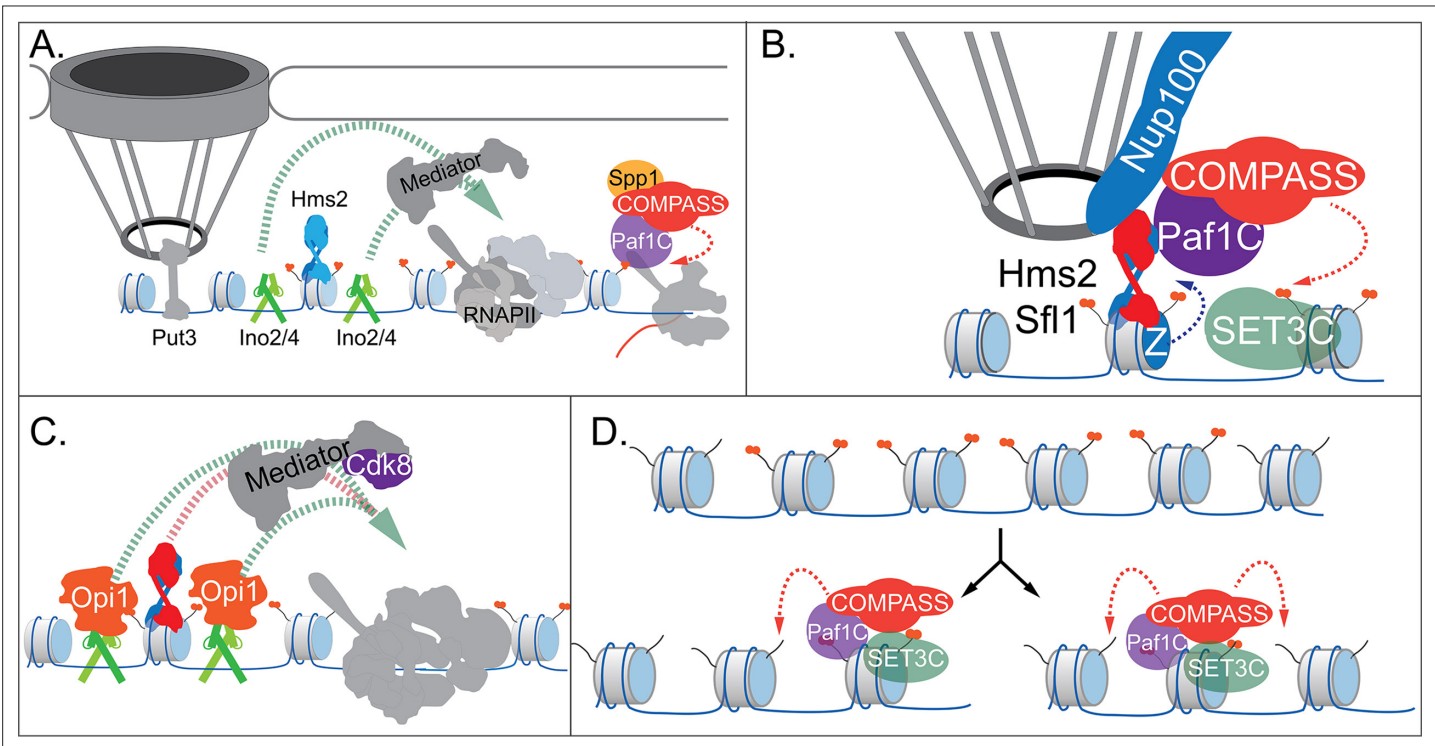

**Figure 7.** Models for *INO1* transcriptional memory. (**A**). *INO1* under activating conditions. Interaction with the nuclear pore complex (NPC) is mediated by Put3 (and, potentially, Cbf1; *Ahmed et al., 2010*; *Randise-Hinchliff et al., 2016*). Ino2/Ino4 heterodimers bind UAS_INO elements flanking the memory recruitment sequence (MRS) and recruit coactivators such as Mediator to promote RNA polymerase II (RNAPII) recruitment and transcription. Hms2 binds to the MRS element but does not contribute to *INO1* transcription or localization. (**B**) Upon repression, *INO1* transcriptional memory is established by Hms2-dependent recruitment of Sfl1, leading to Nup100-dependent interaction with the NPC. Nup100, Sfl1, and Hms2 are required for both Spp1⁻ COMPASS-dependent H3K4 dimethylation and H2A.Z incorporation near the MRS. SET3C associates with H3K4me2 and is required for its persistence. (**C**) During memory, Cdk8⁺ Mediator is recruited to the *INO1* promoter. This requires Sfl1/Hms2 but may also require other TFs such as Ino2 and Ino4, despite their repression by Opi1. (**D**) H3K4me2 inheritance after DNA replication. Following DNA replication, only half of the nucleosomes bear H3K4me2. SET3C recognizes the H3K4me2-marked nucleosomes and recruits Spp1⁻ COMPASS, which methylates adjacent nucleosomes. In the presence of Sfl1/Hms2, this re-establishment will likely emanate from the MRS outwards.

efficiency of such an inheritance mechanism should increase with the number of nucleosomes (**Cutter DiPiazza et al., 2021**). If so, then the duration of memory could be limited by a combination of the number of modified nucleosomes, the fidelity, and efficiency of the reader-writer system and the regulation of the initiating transcription factors.

Our current model for *INO1* transcriptional memory is shown in **Figure 7**. When active, *INO1* interacts with the NPC through upstream DNA zip codes and the TFs Put3 and Cbf1. Two TFs essential for transcription, Ino2 and Ino4, bind to two UAS$_{INO}$ elements that flank the MRS. These TFs recruit co-activators, including histone acetyltransferases like SAGA. Hms2 binds the MRS but has no known role in localization or transcription of active *INO1*. RNAPII recruits the Paf1C and COMPASS, leading to H3K4me3 (**Figure 7A**).

Upon repression, the Opi1 repressor binds to Ino2 (**Brickner and Walter, 2004**; **Heyken et al., 2005**), repressing transcription by recruiting the Rpd3L histone deacetylase (**Kadosh and Struhl, 1997**; **Randise-Hinchliff et al., 2016**). To establish memory, Hms2 facilitates recruitment of Sfl1 to the MRS and Nup100-dependent interaction with the NPC (**Figure 7**). Because Nup98 physically interacts with the H3K4 methyltransferases Trithorax in flies (**Pascual-Garcia et al., 2014**) and MLL in mammals (**Franks et al., 2017**), we envision that Nup100 helps recruit Spp1⁻ COMPASS to dimethylate H3K4 and establish memory (**Figure 7**). To-date, we have not observed co-immunoprecipitation of Nup100 with COMPASS (not shown). However, it is possible that Sfl1, Hms2, or Nup100 compete with Spp1 for binding to COMPASS to ensure this switch. Alternatively, these proteins may recruit Paf1C and COMPASS through interaction with Leo1.

Importantly, the H3K4me2 mark is required for both Sfl1 binding and H2A.Z incorporation during memory, forming a positive feedback loop between interaction with the NPC and chromatin changes. Because the MRS at an ectopic site is sufficient to induce chromatin changes without recruiting RNAPII (**Light et al., 2013**) or Mediator (our unpublished results), *cis*-acting promoter elements presumably facilitate RNAPII recruitment. We hypothesize that Ino2/Ino4 collaborate with Sfl1/Hms2 to recruit Cdk8⁺ Mediator, which is essential for RNAPII poising (**Figure 7**). This would explain why loss of H3K4 methylation leads to loss of Sfl1 binding, peripheral localization, and RNAPII.

Following DNA replication, we proposed that inheritance requires methylation of unmodified H3K4 on nucleosomes near those that were previously modified (**Figure 7**; **Moazed, 2011**). This pathway would involve recognition of the H3K4me2 mark on reincorporated nucleosomes by SET3C, recruitment of Spp1⁻ COMPASS, and dimethylation of H3K4 on adjacent nucleosomes. Once the *INO1* promoter nucleosomes are methylated, Sfl1/Hms2 can bind, mediating interaction with the NPC and re-establishing memory. Loss of Leo1 seems to both weaken the interaction between COMPASS and SET3C and to disrupt persistence/inheritance of *INO1* memory, suggesting that it facilitates Spp1⁻ COMPASS recruitment after DNA replication.

Do all genes that exhibit transcriptional memory utilize the same molecular mechanism? No. Several of the molecules required for *INO1* memory are not generally involved in memory, such as Sfl1 and Hms2 (our unpublished data). However, *INO1* epigenetic memory has also identified factors and mechanisms that are implicated in memory more generally and suggest a core, conserved, epigenetic poising mechanism. For example, Nup100-dependent interaction with the NPC is associated with transcriptional memory of several yeast genes and Nup98 plays an essential role in both interferon gamma memory in HeLa cells (**Light et al., 2010**) and in ecdysone memory in flies (**Gozalo et al., 2020**; **Pascual-Garcia et al., 2017**). In yeast and mammals, memory leads to H3K4me2, RNAPII binding, and Cdk8 association (**D'Urso et al., 2016**; **Light et al., 2013**). Therefore, many genes from yeast to mammals employ a common transcriptional poising mechanism. What selects which genes exhibit memory? Many yeast TFs can mediate Nup100-dependent interaction with the NPC (**Brickner et al., 2019**). Therefore, different stimuli may induce memory through regulating the activity of these TFs to induce memory in different subsets of genes. Diverse, transient signals could be interpreted through TFs to alter future fitness in distinct ways for several generations.

## Materials and methods
### Yeast strains
Yeast strains used in this study are listed in **Supplementary file 1**. Strains used in Co-IP were built using the GFP library and TAP tag library from Open Biosystems (**Ghaemmaghami et al., 2003**; **Huh**

*et al., 2003*). The *mrs* mutations used in *Figures 1 and 4* were created at the endogenous locus via homologous recombination as previously described (*Light et al., 2010*). Competition strains were created by making a SNP mutation in integrative plasmids using inverse PCR (see below). Insertion of the 11 bp MRS at the *URA3* locus (*Figure 6*) was done as previously described (*Light et al., 2010*).

## Chemicals and reagents

All chemicals, except those noted otherwise were obtained from Sigma Aldrich (St. Louis, MO). Yeast media components were from Sunrise Science (Knoxville, TN) and Genesee Scientific (San Diego, CA). Oligonucleotides (listed in *Supplementary file 2*) are from Integrated DNA Technologies (Skokie, IL), and restriction enzymes from New England Biolabs (Woburn, MA). Antibodies used in ChIP experiments: M-280 Sheep α-Mouse IgG and M-280 Sheep α-Rabbit IgG, from Thermo Fisher Scientific; α-H3K4me2 (ab32356), α-GFP (ab290), α-Myc (ab32), and α-H2A.Z (ab4174) from Abcam; α-RNA Polymerase II (cat:664906) from BioLegend; α-TAP (Product #CAB1001) from Invitrogen. The α-GFP nanobody plasmid was a generous gift from Professor Michael Rout (Rockefeller University). The nanobody was expressed, purified, and conjugated to magnetic Dynabeads as described (*Fridy et al., 2014*). The α-H3K4me3 antibody for ChIP was a generous gift from Dr. Ali Shilatifard (Northwestern University).

## Chromatin localization assay

Chromatin localization was performed as described (*D'Urso et al., 2016*; *Egecioglu et al., 2014*), using the confocal SP8 microscope in the Northwestern University Biological Imaging Facility. Error bars represent the SEM of three biological replicates of ≥30 cells. Biological replicates are from separate yeast cultures.

## Reverse transcriptase real time quantitative PCR (RT-qPCR)

For experiments in which mRNA levels were quantified, RT-qPCR was performed as described (*Brickner et al., 2007*). cDNA recovered from RT was analyzed by qPCR using primers found in *Supplementary file 2*, and *INO1* mRNA was normalized to *ACT1* mRNA. Error bars represent the SEM of three biological replicates. Biological replicates are samples from separate yeast cultures.

## Sequencing-based competition assay

For competition experiments, a single mutation was introduced in the pRS306 plasmid (*Sikorski and Hieter, 1989*) by inverse PCR using primers found in *Supplementary file 2*. This plasmid, or the WT ancestral plasmid, was integrated at the *URA3* locus in pairs of strains to be competed. For *Figure 1F*, the two strains (one carrying the SNP 'C' and the other carrying the WT 'A') were combined at various ratios (e.g. 0, 5, 25, 40, 50, 60, 75, 90, and 100% 'A'), estimated using O.D.$_{600}$. Genomic DNA was prepared from mixed populations as described (*Zentner et al., 2015*) and PCR amplified around the SNP region, primers for which are found in *Supplementary file 2*. The resulting chromatograms were analyzed by quantifying the area under the curve of the A/C SNP peak using a custom R script (https://github.com/jasonbrickner/SeqComp; *Brickner, 2022*). For the competition experiments, the strains would be combined at a 1:1 ratio at the start of the competition (as measured by O.D.$_{600}$) and the resulting peak ratio values would be used to normalize the change in peak levels between 0 hr and 3 hr of competition. Additionally, in *Figure 5D*, the two SNPs used to quantify the abundance of the strains were from the *SSN3* and *ssn3-as* mutation (Y236G), amplified and sequenced using primers in *Supplementary file 2*. Error bars throughout represent the SEM of at least three biological replicates. Biological replicates are from separate yeast cultures.

## Chromatin immunoprecipitation

ChIP was performed as previously described (*D'Urso et al., 2016*; *Egecioglu et al., 2014*). Primary rabbit antibodies (anti-H3K4me2, anti-H3K4me3, anti-GFP, and anti-H2A.Z) were recovered with Sheep anti-Rabbit magnetic bead antibodies. Primary mouse antibodies (anti-RNAPII and anti-Myc) were recovered with Sheep anti-Mouse antibodies. DNA recovered from ChIP experiments was analyzed by qPCR using primers in *Supplementary file 2*, using TaqMan qPCR for either the *INO1* promoter or the *PRM1* coding sequence. Error bars represent the SEM of three biological replicates. Biological replicates are defined as samples started from separate yeast cultures.

## Co-immunoprecipitation

Co-immunoprecipitation was performed as previously described (*Gerace and Moazed, 2014*). When using the GFP nanobodies (described above) to pull down Set3-GFP, Spp1-GFP, and Nup100-GFP, 2 mg of proteins in 1 ml was used. The input and immunoprecipitated (IP) fractions (after three washes and 10 min at 65°C in sodium dodecylsulfate sample buffer) were analyzed by immunoblotting using rabbit anti-TAP primary antibodies and Goat anti-Rabbit horseradish peroxidase secondary antibodies.

## Immunoblotting

Samples were separated on 10% NuPAGE Bis-Tris gels in MES running buffer (Invitrogen), transferred to nitrocellulose, analyzed for total protein loaded with Ponceau S stain (Boston BioProducts), blocked with 2% milk for 30 min, and incubated with rabbit anti-TAP primary antibody (Invitrogen) overnight at 4°C. Blots were then washed three times with tris buffered saline (TBS), incubated with Goat anti-Rabbit HRP secondary antibody for 1 hr, washed three times with TBS + 0.1% tween-20, then exposed to enhanced chemiluminescence reagents (Pierce) for 5 min and imaged on an Azure C600 Gel Imaging System.

## Acknowledgements

The authors thank Professor Michael Rout (Rockefeller University) for generously sharing constructs and protocols for expression, purification, and conjugation of the anti-GFP nanobody, Professor Ali Shilatifard (Northwestern University) for generously sharing the anti-H3K4me3 antibody, and members of the Brickner laboratory for helpful comments in this manuscript. AD was supported by the Cell and Molecular Basis of Disease T32 training grant (GM008061). This work was also supported by R01 GM118712 (JHB) and R35 GM136419 (JHB).

## Additional information

### Competing interests

Agustina D'Urso: is affiliated with Arcturus Therapeutics. The author has no financial interests to declare. The other authors declare that no competing interests exist.

### Funding

| Funder | Grant reference number | Author |
| --- | --- | --- |
| National Institute of General Medical Sciences | R35 GM136419 | Bethany Sump<br>Donna G Brickner<br>Agustina D'Urso<br>Seo Hyun Kim<br>Jason H Brickner |
| National Institute of General Medical Sciences | R01 GM118712 | Bethany Sump<br>Donna G Brickner<br>Agustina D'Urso<br>Seo Hyun Kim<br>Jason H Brickner |
| National Institute of General Medical Sciences | T32 GM008061 | Agustina D'Urso |

The funders had no role in study design, data collection and interpretation, or the decision to submit the work for publication.

### Author contributions

Bethany Sump, Conceptualization, Data curation, Investigation, Methodology, Validation, Visualization, Writing - original draft, Writing - review and editing; Donna G Brickner, Data curation, Formal analysis, Investigation; Agustina D'Urso, Conceptualization, Investigation; Seo Hyun Kim, Investigation, Methodology; Jason H Brickner, Conceptualization, Formal analysis, Funding acquisition, Project administration, Supervision, Visualization, Writing - review and editing

**Author ORCIDs**
Jason H Brickner http://orcid.org/0000-0001-8019-3743

**Decision letter and Author response**
Decision letter https://doi.org/10.7554/eLife.77646.sa1
Author response https://doi.org/10.7554/eLife.77646.sa2

## Additional files

### Supplementary files
- Transparent reporting form
- Supplementary file 1. Yeast strains.
- Supplementary file 2. Oligonucleotides.

### Data availability
The scripts used to analyze the competition experiments are available at https://github.com/jason-brickner/SeqComp, (copy archived at swh:1:rev:3eb152971adc09d0b4559bb86e4ad036eebd4dc7).

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
