## [Editor Report]

The findings in this report are highly significant in providing evidence that a positive feedback loop exists between Sfl1-dependent interaction with the nuclear pore complex and H3K4me2 deposition in the INO1 promoter, which is essential for recruitment of poised Pol II, but which does not require transcription initiation to occur. This distinguishes this activity of COMPASS from its conventional role in H3K4 methylation that is dependent on transcription. The Pol II-independent mechanism was also shown to require Nup100, SET3C, and the Leo1 subunit of the Paf1 complex. It is further noteworthy that this specialized H3K4me2 deposition can persist through multiple cell cycles, in a manner that appears to be enhanced by physical association between the writer of this mark, the memory-specific form of COMPASS lacking subunit Spp1, and the reader SET3C, in the manner expected for epigenetic spreading of histone methylation.

---

## [Decision Letter]

**Decision letter after peer review:**

Thank you for submitting your article "Mitotically heritable, RNA polymerase II-independent H3K4 dimethylation stimulates INO1 transcriptional memory" for consideration by *eLife*. Your article has been reviewed by 3 peer reviewers, one of whom is a member of our Board of Reviewing Editors, and the evaluation has been overseen by Kevin Struhl as the Senior Editor. The reviewers have opted to remain anonymous.

Essential revisions:

The referees are in agreement that the paper is potentially a valuable contribution and that the finding of transcription independent deposition of H3K4me2 to establish memory, and its epigenetic inheritance, are both of great interest. At the same time, they each raised issues with certain experiments or interpretations that will likely require some additional experimental work, in addition to text revisions, to address, as follows:

1) Test whether deletion of SET1 or introducing the H3K4R substitution upon anchor away of Opi1 impairs memory, to eliminate an alternative model that Opi1 mediated repression is the primary driver of memory.

2) Try to clarify why nuclear depletion of Opi1 is required to observe that memory stimulates the rate of INO1 derepression during reactivation; and if this is not the case in cells containing Opi1, how can you account for the increased fitness conferred by memory in WT cells?

3) Provide additional qRT-PCR measurements of INO1 mRNA levels to confirm their changes in key TF mutants. The experiment on deleting the TATA box in Figure 1 is a case in point, where only localization of Nup1 was examined. For this experiment, it could be shown instead that H3K4Me2 deposition is unaffected by the TATA deletion.

4) The coIP experiment in Figure 6 does not provide compelling support for the read-write model of H3K4me2 propagation and is merely consistent with it. The conceptual problem is that not all genes behave like INO1, as a result of which the coIP is unlikely to reflect how complexes at one particular locus in the genome are changing. If the authors want to generalize their conclusions, there needs to be more genome-wide data. Another possible way to address this (although it is a much weaker experiment) is to do the same coIP in different deletion backgrounds where cells exhibit the same phenotype as the leo1 deletion.

5) Regarding this last point, it seems important to extend the analysis in Figure 6D to show that the persistence of H3K4me2 deposition following depletion of Sfl1 will be diminished in cells lacking Set3 to bolster the model that the proposed epigenetic inheritance of H3K4me2 requires Set3C as a reader of this histone mark.

6) It would enhance the scientific quality of the work if it was shown that Nup100 mediates establishment or maintenance of memory or both?

7) In addition to performing these experiments, the authors are asked to either conduct other new experiments or make the appropriate revisions of text needed to address all of the other comments raised in the reviews, which could entail softening their conclusions or acknowledging that additional work will be required to address the relevant issues.

*Reviewer #1 (Recommendations for the authors):*

– Additional discussion is required to explain why nuclear depletion of Opi1 is required to observe that memory stimulates the rate of INO1 derepression during reactivation; and if this is not the case in WT cells, how can they account for the increased fitness conferred by memory?

– It seems important to extend the analysis in Figure 6D to show that the persistence of H3K4me2 deposition following depletion of Sfl1 will be diminished in cells lacking Set3 to bolster the model that the proposed epigenetic inheritance of H3K4me2 requires Set3C as a reader of this histone mark.

*Reviewer #2 (Recommendations for the authors):*

1. In Figure 1, the authors show that mutations in the TATA box of the INO1 promoter or mutations in the RNAP II Rbp1 subunit do not compromise transcriptional memory, demonstrating that transcription itself does not appear to be required for memory. However, the readout for memory here is localization of the tagged locus to the nuclear periphery. It would further solidify the authors' conclusions to assess additional transcriptional memory marks. Can authors show that H3K4Me2 or potentially other aspects like H2AZ incorporation also do not get affected?

2. The Leo1 finding is very interesting. To further corroborate their conclusions on the role of Leo1 in maintenance of memory and H3K4Me2, but not in its establishment, authors could assess the timing of recruitment of Leo1 to INO1 – does Leo1 get recruited later on in the memory process? Does it stay at the INO1 promoter as long as the H3K4me2 mark persists? Can it be detected at the ectopically inserted MRS element at the URA3 locus?

3. Sfl1 was found to act in the opposite manner to Leo1 – using varying times of Sfl1 degradation, authors show that it is required for establishment of H3K4Me2 during early memory but is dispensable for later maintenance. Does the other key factor of this process, Nup100, mediate establishment or maintenance or both? Can it be tested in similar time scales?

*Reviewer #3 (Recommendations for the authors):*

Overall summary of the manuscript

In general, I am very enthusiastic about the system, the anchor away approaches that are used to demonstrate that memory indeed is encoded in the INO1 re-activation process (Figure 1) and that there is a source of RNA polII independent H3K4me2 (Figure 6). These data are the strengths of this manuscript. I am less enthusiastic about some of the more incoherent aspects of the paper which includes conflating transcriptional memory with spatial relocalization (lacO based imaging), interpreting genetic interaction data as biochemical interactions, and finally a read-write model of H3K4me2 propagation. However, the data on a source of H3K4me2 that is transcription independent is compelling and what is particularly unique about the observation is the elegant genetics that makes it possible to capture its transient presence within the 'memory' paradigm. I am less convinced that these modifications themselves are required for re-activation, but I fully support such ideas being a part of the Discussion section. In general, strengthening the manuscript with additional data such as qRT-PCR experiments and removing claims of a read-write model for the propagation of H3K4me2 would make this manuscript a valuable addition to our understanding of the intriguing chromatin dependent processes that maintain epigenetic memory.

Specific comments

Related to Figure 1

The authors must consider experiments where -ino to +ino to -ino is performed using different delay times. The timepoint they are using for re-activation measurements are capturing H3K4me2 levels that are coincident with cells moving to the -ino condition. If this has been already done in an earlier study, it would be appropriate to discuss more carefully how they chose the 3-hour time point to investigate memory by returning cells to the -ino state. The data that would help their case is not the memory of spatial localization but supporting data in the form of either H3K4me2 ChIP-qPCR experiments or qRT-PCR measurements.

The Opi1 experiment is a very well thought and clearly the anchor away experiment reveals that the system does indeed exhibit stronger memory that is obscured by the repressive effects of Opi1. This is not a question about semantics but as presented, the authors seem to be looking at the loss of repression as opposed to faster re-activation. This is concerning since an alternative model emerges wherein the combination of activators and repressors (the authors highlighted ~ 6 of them- Cbf1, Put3, Sfi1, Opi1, Mga1, Hms2) could enable the types of persistent memory responses observed in their experiments. Discussing this as an alternative model would be helpful. A full mathematical model would be an appropriate next step but it is understandably outside the scope of this manuscript.

Error in labeling the memory scheme in Figure 1D (should be -ino1 based on the authors description in the text).

How do H3K4me2 levels changes in the Opi1 anchor away paradigm? This would be a vital addition to a revised version of their manuscript where the authors might reasonably expect an increase in H3K4me2 levels upon an increase in the magnitude of re-activation (or loss of repression).

In general, a major caveat to the authors interpretation is that there is no data mapping a quantitative relationship between the magnitude of re-activation and H3K4me2 levels although the Opi1 system represents an opportunity for such a measurement.

Have the authors tested set1 deletion or H3K4R strains in cells where they can anchor away Opi1? This would resolve the alternative model that Opi1 mediated repression primarily contributes to memory (or lack thereof). If re-activation (or memory) fails to be present this would support their model of an alternative transcription independent source of H3K4me2 being important.

In supplementary figure 1, the authors visualize the INO1 locus tethered to the nuclear periphery with nocodazole dependent inhibition of cell division. The authors claim ‘memory’ persists but I do not understand if the memory they refer to is the spatial relocalization (as their data supports) or transcriptional memory (as their data does not support).

Related to Figure 3

I was struck by the tradeoffs in binding and occupancy between the different TFs and think the data where different TFs have distinct effects on memory/activation is very interesting. However, this data once again compels me to ask the extent to which TF occupancy itself confers memory. How does the overexpression of Hms2 and Mga1 affect memory and re-activation?

While there is an impressive amount of panels of ChIP and imaging measurements, I am fundamentally missing a simple qRT-PCR measurement of INO1 RNA levels and how these levels are affected in the different TF deletion mutants. This is especially important in understanding the exact roles of Hms2 and/or Mga1 and the consequence of the increase RNA polII occupancy in ChIP measurements.

Related to Figure 4

Specifically Figure 4E is designed in a manner where one assumes that the authors are testing whether ssn3-AID (+auxin) and NaPP1 have an additive effect. It seems like they do not. In any event this experiment needs a control where the authors solely test the depletion of ssn3-AID (+auxin). If the effect isn’t additive how does this data affect their model and interpretations of the role of the paused RNA polII complex?

The results of the SPT15 depletion measurements are surprising. How is H3K4me2 present in the memory paradigm especially if it was absent during the activation phase? Perhaps one explanation (which I assume the authors favor) is that there is a transcription independent mechanism of H3K4me2 deposition. I am surprised by the fact that this pathway is as efficient as the transcription dependent one (comparing %IP levels). Either the authors are overestimating the effects, or it is a timing issue of when the authors choose to test re-activation? This conclusion as it relates to Figure 4C must be addressed with either additional experiments or a plausible explanation that accounts for the transcription independent pathway being as efficient as the transcription independent one.

Related to Figure 5

The authors discovered that Leo1 affects memory. While the spatial positioning of INO1 data exhibits memory, the authors must include qRT-PCR data given that memory associated with transcription and its correspondence with fitness are the centerpiece of this manuscript. This critique holds true for experiments in the Main and Supplementary Figures.

Related to Figure 6

Although I could infer the experimental details and the experimental design, I found figure 6A to be a very confusing schematic. It would be helpful to readers to revise this panel.

Figure 6B needs supporting qRT-PCR data to test how the timing of auxin addition affects INO1 expression.

Figure 6C (and Figure 1) are very compelling pieces of data and strongly argue for a transcription or at least an RNA polII independent mode of H3K4me2 deposition. This was surprising and I want to highlight that at least to this reviewer, it was a very exciting finding.

The coIP experiment as designed in Figure 6 provides a weak foundation to propose a read-write model of H3K4me2 propagation. The conceptual problem is that not all genes behave like INO1 as a result of which the coIP is unlikely to really reflect how complexes at one particular locus in the genome are changing. If the authors want to generalize their conclusions, there needs to be more genome wide data. Another possible way to address this (although it is a much weaker experiment) is to do the same coIP in different deletion backgrounds where cells exhibit the same phenotype as leo1 δ.

The authors also must show data with point mutants where "reading" activity has been impaired to preserve both complex stoichiometry and disable only its ability to autonomously propagate an epigenetic signal. In the absence of such data, the manuscript will be better served by removing said data from the manuscript and refocusing it on the novel source of H3K4me2 which is RNA polII independent but Nup100 (and other factors) dependent.

Other overall comments as it relates to the manuscript text

Examples where genetic interaction data is conflated with a biochemical model of protein-protein or protein-DNA interactions. This is a recurring theme in the manuscript. I think the genetic data is very compelling and the manuscript simply needs to express the results as such.

"In other words while Sfi1 is required for the interaction with the NPC and chromatin modifications, interaction with the NPC and chromatin modification are also required for Sfi1 binding"

"Thus our current model is that Hms2 prevents Sfi1 binding to active INO1 promoter but is required for Sfi1 binding during memory, perhaps binding as a heterodimer"

In all plots showing INO1/ACT1 mRNA:

y axis for qRT-PCR suggests absolute mRNA counts; should be relative abundance where the authors are in fact measuring a fold-change.

---

## [Author Response]

Essential revisions:The referees are in agreement that the paper is potentially a valuable contribution and that the finding of transcription independent deposition of H3K4me2 to establish memory, and its epigenetic inheritance, are both of great interest. At the same time, they each raised issues with certain experiments or interpretations that will likely require some additional experimental work, in addition to text revisions, to address, as follows:1) Test whether deletion of SET1 or introducing the H3K4R substitution upon anchor away of Opi1 impairs memory, to eliminate an alternative model that Opi1 mediated repression is the primary driver of memory.

This is a good suggestion. We have previously shown that loss of Set1/COMPASS (Light et al., 2013, D’Urso et al., 2016) or the H3K4R/H3K4A mutations (D’Urso et al., 2016) disrupt INO1 transcriptional memory. We have added a sentence to the introduction to highlight this work. Here, we have added RT qPCR data from the set1∆ strain using the Opi1 Anchor Away system (Figure 2A) to confirm that faster reactivation of INO1 in this strain requires Set1.

2) Try to clarify why nuclear depletion of Opi1 is required to observe that memory stimulates the rate of INO1 derepression during reactivation; and if this is not the case in cells containing Opi1, how can you account for the increased fitness conferred by memory in WT cells?

This is a good point and we apologize for not addressing this more explicitly in the original manuscript. We have added the following to the Results section (along with Figure 1 – Supplement 2):

“Because INO1 reactivation is not obviously faster than activation under these conditions, it seems likely that the fitness difference is due to the rate of reactivation of multiple inositol-regulated genes, including INO1. Consistent with this idea, we find that CHO1 (encoding phosphatidyl serine synthase), which is also repressed by inositol, also shows sustained, Nup100-dependent peripheral localization for at least 3h after repression, H3K4me2 and RNAPII binding after repression and faster reactivation (Figure 1 – Supplement 2). This protein is required for growth in the absence of inositol (Atkinson et al., 1980). Therefore, inositol memory impacts other genes as well and the fitness difference is likely due to the coordinated, enhanced reactivation of a set of genes that promote adaptation to inositol starvation.”

3) Provide additional qRT-PCR measurements of INO1 mRNA levels to confirm their changes in key TF mutants. The experiment on deleting the TATA box in Figure 1 is a case in point, where only localization of Nup1 was examined. For this experiment, it could be shown instead that H3K4Me2 deposition is unaffected by the TATA deletion.

Good suggestions. Previous work has shown that sfl1∆ mutants show a defect in the rate of reactivation (D’Urso et al., 2016), which we highlight in the text. Also, we have now added qRT PCR for the hms2∆ strain activated or reactivated using the Opi1 Anchor Away (Figure 3C), confirming that Hms2 is required for faster reactivation in this system.

To confirm that previous transcription is not required for H3K4 methylation during memory, we performed ChIP against H3K4me2 and H3K4me3 in the wild type and tata mutant strains under repressing and memory conditions, as suggested. Consistent with the localization data, we find normal H3K4me2 under memory conditions in the tata mutant strain (Figure 4E).

4) The coIP experiment in Figure 6 does not provide compelling support for the read-write model of H3K4me2 propagation and is merely consistent with it. The conceptual problem is that not all genes behave like INO1, as a result of which the coIP is unlikely to reflect how complexes at one particular locus in the genome are changing. If the authors want to generalize their conclusions, there needs to be more genome-wide data. Another possible way to address this (although it is a much weaker experiment) is to do the same coIP in different deletion backgrounds where cells exhibit the same phenotype as the leo1 deletion.

We have attempted to be appropriately conservative in our interpretation. Our data support a physical interaction between COMPASS and SET3C. While we believe this interaction is important for inheritance of inositol memory, as measured by H3K4me2 of the INO1 promoter, it is possible that it is also important for H3K4me2 at other sites around the genome under a variety of conditions. This may account for the prevalence of the interaction. If this were true, we would expect that the Spp1^-^ form of COMPASS would specifically participate in this interaction. Consistent with this notion, we did not see CoIP with Set3 when the Spp1 subunit was tagged, instead of the Swd1 subunit (now included in Figure 6F).

5) Regarding this last point, it seems important to extend the analysis in Figure 6D to show that the persistence of H3K4me2 deposition following depletion of Sfl1 will be diminished in cells lacking Set3 to bolster the model that the proposed epigenetic inheritance of H3K4me2 requires Set3C as a reader of this histone mark.

Excellent suggestion. We have now included experiments in which we degraded either Sfl1-AID alone or Sfl1-AID and Set3-AID together and assessed the persistence of H3K4me2 over either the INO1 promoter or at URA3:MRS (Figure 6D and E). Remarkably, both loci showed the same efficiency of reincorporation of H3K4me2 and duration of inheritance when Sfl1 alone was degraded (~6-8h; Figure 6D and E). However, when both Sfl1 and Set3 were degraded, H3K4me2 was rapidly lost (Figure 6D and E).

6) It would enhance the scientific quality of the work if it was shown that Nup100 mediates establishment or maintenance of memory or both?

This is a worthwhile experiment that we have tried unsuccessfully. N-terminal and C-terminal AID alleles of Nup100 are neither perfectly functional nor efficiently degraded. Thus, we have not attempted to analyze them in the same way as we have for Sfl1 and Ssn3.

Because loss of Sfl1 leads to loss of peripheral localization (Figure 6B), it seems likely that Nup100 is critical for establishment of H3K4me2, but not essential for its inheritance through mitosis. To strengthen this connection, we now include additional characterization of the nup100∆ phenotype. ChIP against H3K4me2 over time after repression shows that, unlike the wild type strain in which we observe H3K4me2 for > 6h after repression, H3K4me2 is lost within 1h in nup100∆ (Figure 2D), similar to the effect of this mutation on INO1 gene localization (Figure 3D).

Reviewer #1 (Recommendations for the authors):– Additional discussion is required to explain why nuclear depletion of Opi1 is required to observe that memory stimulates the rate of INO1 derepression during reactivation; and if this is not the case in WT cells, how can they account for the increased fitness conferred by memory?

First, we thank Reviewer 1 for their positive assessment of this work.

Related to the source of the fitness advantage of inositol memory, we have included additional text in the Results and in the Discussion regarding our model for the fitness advantage (described above). Importantly, we also include new data showing that another gene (CHO1) that is important for adaptation to growth without inositol also shows Nup100-dependent localization at the nuclear periphery, H3K4me2 and binding of RNAPII during memory, as well as faster reactivation. This suggests that several genes may be more efficiently and coordinately upregulated during memory and that this promotes adaptation to inositol starvation.

– It seems important to extend the analysis in Figure 6D to show that the persistence of H3K4me2 deposition following depletion of Sfl1 will be diminished in cells lacking Set3 to bolster the model that the proposed epigenetic inheritance of H3K4me2 requires Set3C as a reader of this histone mark.

We have degraded either Sfl1-AID alone or Sfl1-AID and Set3-AID together and assessed the persistence of H3K4me2 over either the INO1 promoter or at URA3:MRS (Figure 6D and E). Remarkably, both loci showed the same efficiency of reincorporation of H3K4me2 and duration of inheritance when Sfl1 alone was degraded (~6-8h; Figure 6D and E). However, when both Sfl1 and Set3 were degraded, H3K4me2 was rapidly lost (Figure 6D and E). Thus, the maintenance and inheritance of H3K4me2 requires Set3.

Reviewer #2 (Recommendations for the authors):1. In Figure 1, the authors show that mutations in the TATA box of the INO1 promoter or mutations in the RNAP II Rbp1 subunit do not compromise transcriptional memory, demonstrating that transcription itself does not appear to be required for memory. However, the readout for memory here is localization of the tagged locus to the nuclear periphery. It would further solidify the authors' conclusions to assess additional transcriptional memory marks. Can authors show that H3K4Me2 or potentially other aspects like H2AZ incorporation also do not get affected?

We have included new data in Figure 4E showing that under memory conditions, the INO1 promoter in both the wild type and the tata mutant strains are marked by H3K4me2. This is not observed under repressing conditions and under nether condition is H3K4me3 observed, consistent with this mark being strictly associated with active transcription. Therefore, the H3K4 dimethylation observed under memory conditions is independent of previous transcription.

2. The Leo1 finding is very interesting. To further corroborate their conclusions on the role of Leo1 in maintenance of memory and H3K4Me2, but not in its establishment, authors could assess the timing of recruitment of Leo1 to INO1 – does Leo1 get recruited later on in the memory process? Does it stay at the INO1 promoter as long as the H3K4me2 mark persists? Can it be detected at the ectopically inserted MRS element at the URA3 locus?

The Paf1C complex is required for all H3K4 methylation, suggesting that it is present whenever H3K4 methylation is observed, including during active transcription. Because Leo1 co-purifies with the Paf1C complex, we do not propose that the protein is binding only during memory or that Leo1 is functioning independently of the rest of the complex (Figure 7). Our current model is that Leo1 is present whenever we observe H3K4 methylation. However, we have not shown that it binds by ChIP.

3. Sfl1 was found to act in the opposite manner to Leo1 – using varying times of Sfl1 degradation, authors show that it is required for establishment of H3K4Me2 during early memory but is dispensable for later maintenance. Does the other key factor of this process, Nup100, mediate establishment or maintenance or both? Can it be tested in similar time scales?

This is an excellent suggestion. However, our attempts to make conditional alleles of Nup100 have not been successful to-date. Both amino terminal and carboxyl terminal AID tags show both partial loss of function and are incompletely degraded upon addition of auxin. Therefore, we could not test the role of Nup100 in establishment vs inheritance. However, two observations suggest that Nup100, like Sfl1, is important for establishing H3K4me2 during memory, but not for its inheritance: (1) a null mutation in NUP100 leads to rapid loss of both H3K4me2 (Figure 2D), and peripheral localization (Figure 5E), suggesting a failure to establish memory, (this is distinct from the delayed defect observed in the leo1∆ mutant (Figure 5E)) and (2) degradation of Sfl1 leads to loss of peripheral localization (Figure 6), suggesting loss of the interaction with the nuclear pore complex, but does not lead to rapid loss of H3K4me2.

Reviewer #3 (Recommendations for the authors):Specific commentsRelated to Figure 1The authors must consider experiments where -ino to +ino to -ino is performed using different delay times. The timepoint they are using for re-activation measurements are capturing H3K4me2 levels that are coincident with cells moving to the -ino condition. If this has been already done in an earlier study, it would be appropriate to discuss more carefully how they chose the 3-hour time point to investigate memory by returning cells to the -ino state. The data that would help their case is not the memory of spatial localization but supporting data in the form of either H3K4me2 ChIP-qPCR experiments or qRT-PCR measurements.

We apologize for any lack of clarity that led to confusion on our protocol. All of the ChIP experiments for H3K4me2 and RNAPII were performed after 3h of repression, the same moment when cells were shifted back to -ino medium in competition or qRT PCR experiments and the same time point at which rapamycin was added for the anchor away experiments. However, to confirm that H3K4me2 is associated with the full duration of memory, we performed ChIP in cells shifted to repressing conditions for 0-6h (Figure 2D). This experiment confirmed that, like INO1 localization (Brickner et al., 2007) and RNAPII binding (Light et al., 2013), H3K4me2 persists for > 6h after transcription has been repressed.

The Opi1 experiment is a very well thought and clearly the anchor away experiment reveals that the system does indeed exhibit stronger memory that is obscured by the repressive effects of Opi1. This is not a question about semantics but as presented, the authors seem to be looking at the loss of repression as opposed to faster re-activation. This is concerning since an alternative model emerges wherein the combination of activators and repressors (the authors highlighted ~ 6 of them- Cbf1, Put3, Sfi1, Opi1, Mga1, Hms2) could enable the types of persistent memory responses observed in their experiments. Discussing this as an alternative model would be helpful. A full mathematical model would be an appropriate next step but it is understandably outside the scope of this manuscript.

This is a good point and something we will consider in future studies. In Figure 7, we have attempted to summarize our current understanding as well as providing hypotheses to be tested in the future. Modeling may require a better molecular understanding of how these TFs are regulated and how they recruit critical co-factors. Sfl1 and Hms2 have poorly understood roles as activators and have also been described as repressors. Our current model is that they play a dual role in memory: both altering chromatin structure and facilitating recruitment of Cdk8^+^ Mediator, perhaps in collaboration with Ino2/4. The outcome is a poised state that prevents both Opi1-mediated repression and active transcription. Consistent with this model, loss of upstream factors like Sfl1 and Hms2 leads to full Opi1 repression, while loss of Cdk8 kinase activity leads to loss of the poised RNAPII PIC, but maintenance of H3K4me2.

Error in labeling the memory scheme in Figure 1D (should be -ino1 based on the authors description in the text)

Actually, this label is correct. In the Opi1 anchor away experiments, activation/de-repression was achieved by addition of rapamycin in the presence of inositol.

How do H3K4me2 levels changes in the Opi1 anchor away paradigm? This would be a vital addition to a revised version of their manuscript where the authors might reasonably expect an increase in H3K4me2 levels upon an increase in the magnitude of re-activation (or loss of repression).

Because memory is established by switching from -inositol to +inositol for 3h when reactivating INO1 using Opi1 anchor away, the H3K4me2 levels are the same as in all of the memory ChIP experiments.

In general, a major caveat to the authors interpretation is that there is no data mapping a quantitative relationship between the magnitude of re-activation and H3K4me2 levels although the Opi1 system represents an opportunity for such a measurement.

We apologize if there was a misunderstanding. The anchor away of Opi1 was only used during the final activation/reactivation time course. Prior to that, memory was established using exactly the same conditions we used throughout the rest of the paper: overnight growth in -inositol, followed by 3h of repression in +inositol. We have clarified this in the Results section with the following text:

“To avoid this possible complication, we induced INO1 transcription by removing Opi1 from the nucleus using the Anchor-Away method (Haruki et al., 2008), either in cells that were grown continuously in the presence of inositol (i.e. activation) or in cells that were grown overnight in the absence of inositol and then shifted into medium containing inositol for 3h (i.e. reactivation; Figure 1C).”

In other words, all of the ChIP experiments against H3K4me2 from non-anchor away strains represent the 0 minute time point for the anchor away experiments.

Have the authors tested set1 deletion or H3K4R strains in cells where they can anchor away Opi1? This would resolve the alternative model that Opi1 mediated repression primarily contributes to memory (or lack thereof). If re-activation (or memory) fails to be present this would support their model of an alternative transcription independent source of H3K4me2 being important.

We now include qRT PCR data for INO1 transcription upon Opi1 anchor away in the set1∆ mutant (Figure 2A), confirming that Set1 is essential for the faster reactivation of INO1 during memory.

In supplementary figure 1, the authors visualize the INO1 locus tethered to the nuclear periphery with nocodazole dependent inhibition of cell division. The authors claim 'memory' persists but I do not understand if the memory they refer to is the spatial relocalization (as their data supports) or transcriptional memory (as their data does not support).

We interpret sustained peripheral localization as reflecting inositol memory. Given that interaction with Sfl1/Hms2 leads to Nup100dependent interaction with the NPC and that these factors are essential for chromatin modifications, binding of RNAPII and faster reactivation upon Opi1 anchor away, peripheral localization is a reasonable proxy for the other molecular events that promote transcriptional memory.

Related to Figure 3I was struck by the tradeoffs in binding and occupancy between the different TFs and think the data where different TFs have distinct effects on memory/activation is very interesting. However, this data once again compels me to ask the extent to which TF occupancy itself confers memory. How does the overexpression of Hms2 and Mga1 affect memory and re-activation?

We have not tested the effects of overexpression of these TFs. While it is a worthwhile thing to try, in case it is sufficient to induce memory, it is unclear if the negative result (i.e. no effect) is informative. We respectfully disagree that such an experiment is essential to establish that Hms2 is important for memory.

While there is an impressive amount of panels of ChIP and imaging measurements, I am fundamentally missing a simple qRT-PCR measurement of INO1 RNA levels and how these levels are affected in the different TF deletion mutants. This is especially important in understanding the exact roles of Hms2 and/or Mga1 and the consequence of the increase RNA polII occupancy in ChIP measurements.

Such a panel has been published previously for sfl1∆ (D’Urso et al., 2016). We now include qRT-PCR of INO1 activation and reactivation using Opi1 anchor away in the hms2∆ mutant (Figure 3C), which confirms that Hms2 plays an essential and specific role in promoting faster reactivation.

Related to Figure 4Specifically Figure 4E is designed in a manner where one assumes that the authors are testing whether ssn3-AID (+auxin) and NaPP1 have an additive effect. It seems like they do not. In any event this experiment needs a control where the authors solely test the depletion of ssn3-AID (+auxin). If the effect isn't additive how does this data affect their model and interpretations of the role of the paused RNA polII complex?

If we understand the concern, we fear we have confused the reviewer. If so, we apologize. The experiments in Figure 4 utilized an analog sensitive allele of Ssn3, not Ssn3-AID. In Figure 4F and H, we combined inhibition of Ssn3 with degradation of Set3-AID. However, we also performed the control of inhibiting Ssn3 without degrading Set3. If the reviewer is asking if inactivating Set3 alone has an effect, the answer is yes. D’Urso et al., (2016) showed that depletion of Set3 by anchor away leads to rapid loss of H3K4me2 during memory, but not under activating conditions.

The results of the SPT15 depletion measurements are surprising. How is H3K4me2 present in the memory paradigm especially if it was absent during the activation phase? Perhaps one explanation (which I assume the authors favor) is that there is a transcription independent mechanism of H3K4me2 deposition. I am surprised by the fact that this pathway is as efficient as the transcription dependent one (comparing %IP levels). Either the authors are overestimating the effects, or it is a timing issue of when the authors choose to test re-activation? This conclusion as it relates to Figure 4C must be addressed with either additional experiments or a plausible explanation that accounts for the transcription independent pathway being as efficient as the transcription independent one.

We conclude from our data that transcription is essential for H3K4 methylation under activating conditions and that it is dispensable under memory conditions. It is unclear to us what the expectation for efficiency for each of these mechanisms is but our ChIP qPCR suggest that they are comparably efficient. To directly address whether previous H3K4 methylation under activating conditions is required for H3K4me2 under memory conditions, we now include ChIP analysis of the ino1-tata mutant, which is not transcribed. We performed ChIP against H3K4me2 and H3K4me3 in the wild type and tata mutant strains under repressing and memory conditions.

Consistent with the localization data, we find normal H3K4me2 under memory conditions in the tata mutant strain (Figure 4E). Therefore, previous RNAPII-dependent H3K4 methylation is not essential for H3K4 dimethylation during memory.

Related to Figure 5The authors discovered that Leo1 affects memory. While the spatial positioning of INO1 data exhibits memory, the authors must include qRT-PCR data given that memory associated with transcription and its correspondence with fitness are the centerpiece of this manuscript. This critique holds true for experiments in the Main and Supplementary Figures.

qRT-PCR for the rate of INO1 activation and reactivation in wild type and leo1∆ mutant strains is shown in Figure 5C. The dependence of the fitness advantage on Leo1 is shown in Figure 5D.

Related to Figure 6Although I could infer the experimental details and the experimental design, I found figure 6A to be a very confusing schematic. It would be helpful to readers to revise this panel.

We have added additional description of this schematic to the legend and included the words “fix for ChIP” beside the bottom arrow of the scheme to clarify this figure.

Figure 6B needs supporting qRT-PCR data to test how the timing of auxin addition affects INO1 expression.

We believe that this experiment is not essential for the interpretation of our results. Loss of Sfl1 leads to a defect in transcription (D’Urso et al., 2016). The loss of RNAPII in these experiments upon Sfl1 degradation argues that Sfl1 is required for the establishment and continuous presence of RNAPII (Figure 7). The important finding of this experiment is that Sfl1 is required for both establishment and maintenance of peripheral localization, H2A.Z and RNAPII binding during memory, but that it is required for establishment, but not maintenance of H3K4me2 during memory. Furthermore, conditional inactivation of Sfl1 once memory is established demonstrated that RNAPII-independent H3K4me2 is mitotically heritable.

Figure 6C (and Figure 1) are very compelling pieces of data and strongly argue for a transcription or at least an RNA polII independent mode of H3K4me2 deposition. This was surprising and I want to highlight that at least to this reviewer, it was a very exciting finding.The coIP experiment as designed in Figure 6 provides a weak foundation to propose a read-write model of H3K4me2 propagation. The conceptual problem is that not all genes behave like INO1 as a result of which the coIP is unlikely to really reflect how complexes at one particular locus in the genome are changing. If the authors want to generalize their conclusions, there needs to be more genome wide data. Another possible way to address this (although it is a much weaker experiment) is to do the same coIP in different deletion backgrounds where cells exhibit the same phenotype as leo1 δ.

We agree that this experiment alone does not settle the mechanism. We have attempted to be appropriately conservative in our interpretation. Our data support a physical interaction between COMPASS and SET3C. Based on that interaction, we propose a model that will guide experiments in the future. While we believe this interaction is important for inheritance of inositol memory, as measured by H3K4me2 of the INO1 promoter, it is possible that it is also important for H3K4me2 at other sites around the genome under a variety of conditions, which may account for the prevalence of the interaction. If this were true, we would expect that the Spp1^-^ form of COMPASS would specifically participate in this interaction. Consistent with this notion, we now include new data in which we did not see CoIP with Set3 when the Spp1 subunit was tagged, instead of the Swd1 subunit (now included in Figure 6F).

The authors also must show data with point mutants where "reading" activity has been impaired to preserve both complex stoichiometry and disable only its ability to autonomously propagate an epigenetic signal. In the absence of such data, the manuscript will be better served by removing said data from the manuscript and refocusing it on the novel source of H3K4me2 which is RNA polII independent but Nup100 (and other factors) dependent.

We have previously published that mutation of tryptophan 140 to alanine, which the Buratowski group showed disrupts interaction of Set3 with H3K4me2, also disrupts INO1 memory (D’Urso et al., 2016). We now highlight this work in the text.

Other overall comments as it relates to the manuscript text.Examples where genetic interaction data is conflated with a biochemical model of protein-protein or protein-DNA interactions. This is a recurring theme in the manuscript. I think the genetic data is very compelling and the manuscript simply needs to express the results as such."In other words while Sfi1 is required for the interaction with the NPC and chromatin modifications, interaction with the NPC and chromatin modification are also required for Sfi1 binding"

Modified to:

“In other words, while Sfl1 is required for interaction of the INO1 gene with the NPC and chromatin modifications, Nup100 and chromatin modification are also required for Sfl1 binding to the INO1 promoter.”

"Thus our current model is that Hms2 prevents Sfi1 binding to active INO1 promoter but is required for Sfi1 binding during memory, perhaps binding as a heterodimer"

Modified to:

“Thus, our genetic results suggest that Hms2 prevents Sfl1 binding to the active INO1 promoter but is required for Sfl1 binding during memory, perhaps binding as a heterodimer.”

In all plots showing INO1/ACT1 mRNA:y axis for qRT-PCR suggests absolute mRNA counts; should be relative abundance where the authors are in fact measuring a fold-change.

We apologize for the confusion. The “INO1/ACT1 mRNA” label as well as the phrase “the INO1 mRNA was quantified relative to ACT1 mRNA by RT-qPCR “ in the legends was meant to indicate that we are plotting the ratio of INO1 to ACT1, measured by qRT PCR. We have also made this statement in the Method